# HYGMA: Hypergraph Coordination Networks with Dynamic Grouping for Multi-Agent Reinforcement Learning

Chiqiang Liu [1]   Dazi Li [1]

## Abstract

Cooperative multi-agent reinforcement learning faces significant challenges in effectively organizing agent relationships and facilitating information exchange, particularly when agents need to adapt their coordination patterns dynamically. This paper presents a novel framework that integrates dynamic spectral clustering with hypergraph neural networks to enable adaptive group formation and efficient information processing in multi-agent systems. The proposed framework dynamically constructs and updates hypergraph structures through spectral clustering on agents' state histories, enabling higher-order relationships to emerge naturally from agent interactions. The hypergraph structure is enhanced with attention mechanisms for selective information processing, providing an expressive and efficient way to model complex agent relationships. This architecture can be implemented in both value-based and policy-based paradigms through a unified objective combining task performance with structural regularization. Extensive experiments on challenging cooperative tasks demonstrate that our method significantly outperforms state-of-the-art approaches in both sample efficiency and final performance. The code is available at: https://github.com/mysteryelder/HYGMA.

## 1. Introduction

Cooperative multi-agent reinforcement learning (MARL) has emerged as a promising paradigm for addressing complex real-world challenges that require coordinated decision-making among multiple agents (Charbonnier et al., 2025; Si et al., 2025). A fundamental challenge in MARL is the efficient organization and coordination of agents to achieve system-wide objectives. This becomes particularly critical when agent relationships need to dynamically evolve in response to changing task demands and environmental conditions.

Despite significant advances in value decomposition methods and policy optimization techniques (Huang et al., 2025; Li et al., 2021), existing MARL approaches face inherent limitations in modeling the dynamic nature of inter-agent relationships. Traditional methods either adopt a uniform treatment of all agents (Tan, 1993; Heess et al., 2017) or rely on static grouping structures (Zhang et al., 2020; Sukhbaatar et al., 2016). Such rigid frameworks often fail to capture the evolving coordination requirements in complex multi-agent systems, leading to suboptimal performance and inefficient information exchange.Recent work has highlighted the importance of adaptive coordination structures in MARL (Zang et al., 2024), yet the development of frameworks that can automatically identify and adapt agent relationships remains an open challenge (Niu et al., 2021).

To address these limitations, this paper proposes HYGMA (HYpergraph Grouping for Multi-Agent coordination), a novel framework that leverages hypergraph networks to capture and adapt the complex relationships in multi-agent systems. Unlike traditional graph-based approaches that can only model pairwise interactions, hypergraph networks naturally represent higher-order relationships among multiple agents (Kim et al., 2024), enabling more expressive and efficient group coordination. The proposed framework dynamically constructs hypergraph structures through spectral clustering on agents' state histories, allowing agent groups to form and evolve based on their actual coordination needs during task execution. The core of our approach is a two-level architecture that separates group formation from information processing. The first level employs dynamic spectral clustering to identify agent groups based on their state histories and interaction patterns. These grouping results are then used to construct hyperedges in the hypergraph network. The second level utilizes hypergraph convolution networks (HGCN) to process and propagate information among agents, enabling efficient feature extraction that respects the identified group structures. This architecture can

---

[1] College of Information Science and Technology, Beijing University of Chemical Technology, Beijing, China. Correspondence to: Dazi Li <lidz@mail.buct.edu.cn>.

*Proceedings of the $42^{nd}$ International Conference on Machine Learning*, Vancouver, Canada. PMLR 267, 2025. Copyright 2025 by the author(s).

be effectively integrated into both value-based and policy-based multi-agent learning frameworks.

The main contributions of this work are summarized as follows:

- A dynamic spectral clustering-based grouping mechanism that adaptively constructs coordination structures through temporal agent state representations, enabling automated group formation and evolution in response to dynamic coordination requirements in multi-agent systems.

- A novel hypergraph neural architecture that extends traditional graph-based information processing to capture higher-order agent relationships, substantially improving the expressiveness and efficiency of multi-agent information exchange through learnable hypergraph convolution operations.

- A unified learning framework that seamlessly incorporates the proposed hypergraph-based coordination mechanism into both value-based and policy-based multi-agent learning paradigms, achieving significant empirical improvements over state-of-the-art approaches in complex cooperative tasks.

## 2. Related Work

**Multi-Agent Coordination Structures** Centralized Training with Decentralized Execution (CTDE) (Oliehoek et al., 2008) has driven significant advances in value decomposition methods. While single-agent settings have explored recursive least-squares approaches (Song et al., 2021), MARL methods like VDN (Sunehag et al., 2017) employed linear summation of local utilities, while QMIX (Rashid et al., 2020) introduced non-linear mixing networks constrained by monotonicity. Subsequent innovations like QTRAN (Son et al., 2019) relaxed these constraints through the Individual-Global-Max principle, with extensions enhancing exploration (Gupta et al., 2021; Mahajan et al., 2019) and mixing capacity (Wang et al., 2020a; Zang et al., 2023). However, these methods fundamentally assume fixed agent relationships through static decomposition structures (e.g., sum/max operations), limiting their ability to model dynamic coordination patterns requiring adaptive group formation. In parallel, actor-critic frameworks have extended CTDE through centralized critics, as seen in MADDPG (Lowe et al., 2017) and COMA (Foerster et al., 2018) which address credit assignment via counterfactual baselines. While such approaches enable decentralized execution, their information processing remains restricted to pairwise interactions. Recent communication-augmented variants (Su et al., 2021; Wan et al., 2021) demonstrate performance gains from explicit information exchange, yet still operate on fixed inter-action graphs that cannot capture the higher-order relationships inherent in complex team coordination.

**Dynamic Grouping Mechanisms** Prior work in agent organization has explored three principal paradigms: predefined grouping (Lhaksmana et al., 2018; Macarthur et al., 2011; Russell & Zimdars, 2003), task-specific allocation (Iqbal et al., 2022; Lee et al., 2019; Shu & Tian, 2018), and role-based decomposition. While role-learning methods like ROMA (Wang et al., 2020b) and RODE (Wang et al., 2020c) enable context-dependent specialization, they typically require manual specification of role numbers or action space partitions. VAST (Phan et al., 2021) demonstrates subgroup-aware value factorization but inherits the limitation of fixed group cardinality. These approaches remain constrained by either domain knowledge requirements or static structural assumptions, struggling to adapt when coordination patterns evolve dynamically during task execution. Recent advances attempt to automate group formation through learned similarity metrics (Zang et al., 2024). However, such methods often rely on heuristic similarity measures or black-box relational modules that lack interpretable temporal analysis. Recent work has also explored coordination graphs for adaptive agent relationships (Yang et al., 2022; Wang et al., 2021). Our spectral clustering approach fundamentally differs by establishing mathematically grounded group discovery through spectral analysis of agents' state trajectory manifolds, enabling principled adaptation to emerging coordination needs without prior group specifications.

**Hypergraph Representation Learning** Graph neural networks (GNNs) (Wang & Gombolay, 2020; Wu et al., 2020) have demonstrated extensive applications in various domains (Jia et al., 2025), and have become prevalent in MARL for modeling agent interactions through graph structures (Pesce & Montana, 2023). Approaches like DGN (Jiang et al., 2018) and MAGNet (Malysheva et al., 2018) employ attention mechanisms to capture pairwise relationships, while dynamic variants (Liu et al., 2020; Sheng et al., 2022) learn adaptive edges through differentiable attention. However, these graph-based methods fundamentally operate on dyadic connections, forcing higher-order group interactions to be approximated through chains of pairwise edges—an inefficient representation that loses natural multi-agent coordination semantics. Addressing coordination complexity, research has advanced along several dimensions: factorization approaches through coordination graphs (Böhmer et al., 2020), regularization techniques via nested optimization (Song et al., 2020), adaptive topological structures (Li et al., 2020), and representational capacity analysis across varied multi-agent domains (Castellini et al., 2021). Policy-based frameworks have incorporated attention mechanisms to enhance selective information processing (Iqbal & Sha, 2019), yet these innovations remain constrained by their underlying pairwise representational paradigm. Even

hierarchical communication architectures (Liu et al., 2023), while enhancing structured information flow, cannot inherently capture the n-ary relationships fundamental to complex team coordination. Hypergraph neural networks (Feng et al., 2019), demonstrating their expressiveness in social network modeling (Yu et al., 2021) and recommendation systems (Gao et al., 2023), offer a theoretical framework capable of natively representing higher-order interactions. To our knowledge, no prior work has combined spectral clustering with hypergraph convolution to dynamically construct multi-agent hyperedges based on temporal state patterns, bypassing the heuristic hyperedge definitions common in other domains.

## 3. Method

A cooperative multi-agent task can be modeled as a Dec-POMDP (Oliehoek et al., 2016) forming the tuple $\langle N, S, A, P, R, \Omega, O, n, \gamma \rangle$, where $N = \{1, ..., n\}$ denotes the finite set of $n$ agents. $s \in S$ represents the true state of the environment from which each agent $i$ draws an individual observation $o_i \in \Omega$ according to the observation function $O(s, i)$. At each timestep, each agent $i$ selects an action $a_i \in A_i$ based on its action-observation history $\tau_i \in T = (\Omega \times A)^*$, forming a joint action $a \in A^n$. This results in a transition to the next state $s'$ according to the transition function $P(s'|s, a) : S \times A \to \Delta(S)$ and a shared reward $r = R(s, a)$ for the team, where $R : S \times A \to \mathbb{R}$ is the reward function and $\gamma \in [0, 1)$ is the discount factor.

The dynamic relationships between agents are modeled through a hypergraph structure (Zhang et al., 2018) $G = (V, E, W)$. The vertex set $V = \{v_1, ..., v_n\}$ represents agents, while the hyperedge set $E = \{e_1, ..., e_m\}$ captures higher-order interactions among multiple agents, where each hyperedge $e_k$ connects a subset of agents. $W \in \mathbb{R}^{m \times m}$ is a diagonal matrix containing hyperedge weights $w_{kk}$ that indicate the strength of these relationships. Unlike traditional graphs limited to pairwise relationships, hypergraphs capture complex group interactions that naturally arise in multi-agent systems. The hypergraph structure updates dynamically through spectral clustering based on agents' state histories, enabling adaptive grouping as the environment evolves.

### 3.1. Overview

The proposed HYGMA framework enhances multi-agent reinforcement learning through dynamic grouping and hypergraph-based information processing. The core components include a dynamic spectral clustering module for adaptive group formation, a hypergraph convolution network with multi-head attention for intra-group information processing. The method has been implemented in both value-based and policy-based learning paradigms. The overall architecture of our proposed method is illustrated in Figure 1.

The dynamic spectral clustering module periodically processes agents' state histories rather than every timestep, identifying natural groupings based on behavioral similarities. These grouping results construct a hypergraph topology where vertices represent agents and hyperedges capture intra-group relationships. Within each group, a multi-layer hypergraph convolution network with multi-head attention mechanism processes information, allowing agents to selectively focus on relevant information rather than simple averaging. For each agent $i$, its local observation $o_i$ and action-observation history $\tau_i$ are encoded through a recurrent neural network to generate individual embeddings, which are then enhanced through the attentive hypergraph convolution layers to produce group-aware representations $h_i$.

In the value-based implementation, each agent's individual Q-value is augmented with its group-aware representation to form $Q_i(\tau_i, a_i, h_i)$. These enhanced Q-values are then combined through a monotonic mixing network to compute the joint action-value $Q_{tot}$. Similarly, in the policy-based implementation, each agent's policy network $\pi_i$ takes the concatenation of local observation history $\tau_i$ and group-aware representation $h_i$ as input to generate action probabilities, while the critic network uses the enhanced joint state representation for value estimation.

Both implementations fully adhere to the CTDE paradigm while enhancing its coordination capabilities. During centralized training, we leverage global information to discover dynamic group structures through spectral clustering. As proven in Theorem 3.2, these group structures converge and stabilize during training. At execution time, the core CTDE principle is preserved—agents operate without access to global state information. The key enhancement is that agents within established groups share local observations with groupmates, enabling the HGCN to generate group-aware representations that capture coordinated behaviors discovered during training. This approach respects CTDE's fundamental requirement of avoiding global state dependency during execution, while allowing more effective coordination through structured local information exchange within stable groups.

### 3.2. Dynamic Spectral Clustering for Group Formation

The group formation process takes agents' state histories as input to identify natural groupings that reflect coordination patterns. Given state history matrix $H_t \in \mathbb{R}^{b \times l \times d}$ for $n$ agents, where $b$ is batch size, $l$ is trajectory length, and $d$ is state dimension, we formalize the grouping as a normalized cut minimization problem:

$$Ncut(G) = \sum_{i=1}^{k} \frac{cut(V_i, \bar{V}_i)}{vol(V_i)} \tag{1}$$

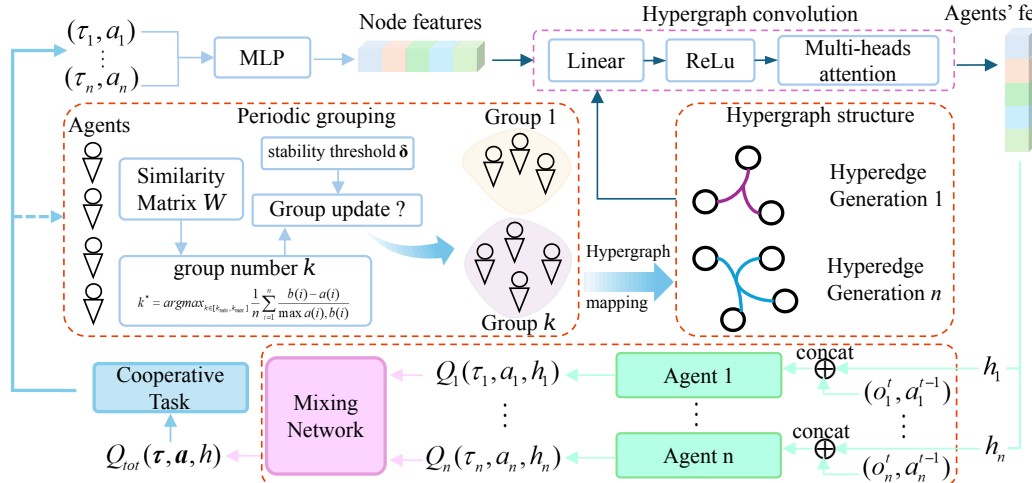

*Figure 1.* Overview of the proposed hypergraph-based dynamic grouping multi-agent reinforcement learning framework with spectral clustering and attentive information processing.

where $cut(V_i, \bar{V}_i)$ measures the sum of edge weights between group $V_i$ and its complement, and $vol(V_i)$ is the sum of degrees of vertices in $V_i$. To solve this NP-hard discrete optimization problem, we adopt a spectral relaxation:

$$\min_{A \in \mathbb{R}^{n \times k}} Tr(A^T L A) \quad s.t. \quad A^T A = I \qquad (2)$$

where $L = D - W$ is the normalized graph Laplacian, with $D$ being the diagonal degree matrix and $W$ representing the similarity matrix constructed using k-nearest neighbors based on Euclidean distances between agents' state history trajectories. Formally, $W_{ij} > 0$ if agent $j$ is among the $k$-nearest neighbors of agent $i$ (or vice versa), and $W_{ij} = 0$ otherwise. State histories are normalized across feature dimensions to ensure consistent scaling in different environments, enabling identification of coordination patterns regardless of specific state representations. The solution $A$ contains the $k$ eigenvectors corresponding to the $k$ smallest eigenvalues of $L$, providing a provably good approximation:

**Theorem 3.1** (Clustering Approximation)**.** *The spectral clustering solution $G$ satisfies:*

$$Ncut(G) \leq O(\log k) \cdot Ncut(G^*) \qquad (3)$$

*where $G^*$ is the optimal grouping structure. Furthermore, this approximation ratio is tight in the sense that no polynomial-time algorithm can achieve asymptotically better approximation under standard complexity assumptions. (See Appendix A.1 for proof)*

The determination of optimal group number $k^*$ requires balancing the expressiveness of grouping structure with its computational efficiency. We address this through optimizing the silhouette score, which measures both the cohesion within groups and the separation between groups:

$$k^* = argmax_{k \in [k_{min}, k_{max}]} \frac{1}{n} \sum_{i=1}^{n} \frac{b(i) - a(i)}{\max a(i), b(i)} \qquad (4)$$

where $a(i)$ represents the mean distance between agent $i$ and all other agents within the same group, and $b(i)$ denotes the minimum mean distance from agent $i$ to any other group. This optimization inherently captures the trade-off between intra-group similarity and inter-group distinctiveness. The dynamic nature of multi-agent interactions necessitates a group update mechanism:

$$G_t = \begin{cases} \mathcal{C}(H_t) & \text{if } \eta(G_{t-1}, \mathcal{C}(H_t)) > \delta \\ G_{t-1} & \text{otherwise} \end{cases} \qquad (5)$$

where $\eta(\cdot, \cdot)$ measures the normalized proportion of agents changing groups, and $\delta$ is a stability threshold. This mechanism guarantees both adaptivity and stability:

**Theorem 3.2** (Convergence)**.** *Under the dynamic update rule with threshold $\delta$, the sequence of groupings $G_t$ converges in finite time with probability 1. The expected number of updates before convergence is bounded by $O(1/\delta)$.(See Appendix A.2 for proof)*

To ensure computational efficiency, we employ optimization strategies including history windowing, periodic updates, and termination.

The quality of grouping structure directly impacts the effectiveness of subsequent learning. For a rigorous characterization of this relationship, let $V^*(s, G)$ denote the optimal value function under grouping structure $G$, we have:

**Theorem 3.3** (Quality Preservation)**.** *For the obtained grouping structure $G_t$ and the optimal grouping $G^*$:*

$$|V^*(s, G^*) - V^*(s, G_t)| \leq \gamma \alpha \log k \qquad (6)$$

*where $\gamma$ is the discount factor and $\alpha$ is a problem-dependent constant. Moreover, this bound is tight when the normalized cut difference between $G_t$ and $G^*$ approaches $O(\log k)$.(See Appendix A.3 for proof)*

These theoretical guarantees establish the effectiveness of the dynamic spectral clustering mechanism in both computational efficiency and learning quality. The resulting group structure forms the foundation for hypergraph-based information processing described in the subsequent section.

### 3.3. Hypergraph Attention Convolution Network

Based on the grouping structure obtained from dynamic spectral clustering, we construct a hypergraph $G = (V, E, W)$ where vertices $V$ represent agents and hyperedges $E$ capture the intra-group relationships. Specifically, for each group identified in Section 3.2, we create a hyperedge connecting all agents within that group, with the edge weight in $W$ reflecting the group cohesion measured by the silhouette score. This dynamic hypergraph construction allows us to model complex, higher-order relationships that naturally emerge from agent interactions. Within this dynamic hypergraph structure, we employ an attention-enhanced hypergraph convolution network to generate personalized information embeddings for each agent. For a node $i$ at layer $l$, the feature update follows:

$$h_i^{(l+1)} = \sigma\left( \sum_{e \in \mathcal{E}i} \alpha_{ie} \cdot D^{-\frac{1}{2}} H W B^{-1} \right.$$
$$\left. \cdot H^T D^{-\frac{1}{2}} h_i^{(l)} P^{(l)} \right) \tag{7}$$

where $\mathcal{E}i$ represents the set of hyperedges containing node $i$, and matrices $H$, $D$, and $B$ define the hypergraph structure. The attention coefficient $\alpha_{ie}$ is computed through:

$$\alpha_{ie} = \frac{\exp(LeakyReLU(a^T[W_s h_i^{(l)} | W_s h_e^{(l)}]))}{\sum_{e' \in \mathcal{E}i} \exp(LeakyReLU(a^T[W_s h_i^{(l)} | W_s h e'^{(l)}]))} \tag{8}$$

This attention mechanism enables each agent to adaptively weight information from different groups it belongs to, generating personalized feature representations that capture both individual characteristics and group dynamics. The processed features enhance both value and policy networks:

$$Q_i(\tau_i, a_i, h_i^{(L)}) = f_Q([\tau_i | a_i | h_i^{(L)}]) \tag{9}$$

$$\pi_i(a_i | \tau_i, h_i^{(L)}) = f_\pi([\tau_i | h_i^{(L)}]) \tag{10}$$

This integration creates an end-to-end trainable architecture where dynamic group structures adaptively influence agents' decision making through attention-weighted information aggregation. By combining dynamic spectral clustering with attention-based hypergraph convolution, HYGMA enables flexible and efficient information exchange within groups while maintaining each agent's ability to selectively process relevant information.

This hypergraph structure not only enhances representation learning but also reduces communication complexity compared to fully-connected architectures, from $O(n^2)$ to $O(n^2/k)$ where $k$ is the number of groups (see Appendix A.4 for theoretical analysis).

### 3.4. Learning Objectives

To optimize HYGMA architecture, we design a joint learning objective that combines the main task performance with group consistency and attention regularization:

$$\mathcal{L}_{\text{total}} = \mathcal{L}_{\text{task}} + \lambda_1 \mathcal{L}_{\text{group}} + \lambda_2 \mathcal{L}_{\text{att}} \tag{11}$$

The group consistency loss $\mathcal{L}_{group}$ ensures effective spectral clustering by minimizing intra-group distances while maximizing inter-group separation:

$$\mathcal{L}_{\text{group}} = \sum_{k=1}^{K} \left( \frac{1}{|C_k|} \sum_{i,j \in C_k} d(h_i, h_j) \right.$$
$$\left. -\beta \min_{l \neq k} \frac{1}{|C_l|} \sum_{i \in C_k, j \in C_l} d(h_i, h_j) \right) \tag{12}$$

To encourage selective information processing through the attention mechanism while avoiding trivial solutions, we employ an entropy regularization term:

$$\mathcal{L}_{\text{att}} = -\sum i = 1^N \sum_{e \in \mathcal{E}i} \alpha_{ie} \log(\alpha_{ie}) \tag{13}$$

The hyperparameters $\lambda_1$ and $\lambda_2$ balance the importance of these auxiliary objectives against the main task learning.

### 3.5. Implementation in Value-based and Policy-based Frameworks

HYGMA framework can be effectively implemented in both value-based and policy-based learning paradigms while maintaining the CTDE principle. In both implementations, we maintain the core components: dynamic spectral clustering for group formation, hypergraph construction based on group structure, and attention-enhanced HGCN for information processing within groups.

#### 3.5.1. VALUE-BASED IMPLEMENTATION

In the QMIX framework, we enhance individual Q-values with group-aware representations:

$$Q_i(\tau_i, a_i, h_i) = f_Q([\tau_i | a_i | h_i]) \tag{14}$$

These enhanced Q-values are then combined through a monotonic mixing network:

$$Q_{tot} = f_{mix}(Q_1, ..., Q_n; s) \tag{15}$$

The TD loss serves as the main task objective:

$$\mathcal{L}_{\text{task}} = (r + \gamma \max a' Q_{tot}(s', a') - Q_{tot})^2 \tag{16}$$

The hypergraph structure particularly enhances the value-based implementation by providing contextual information that helps individual agents understand their role within different coordination patterns. This addresses a fundamental limitation of traditional value factorization methods: while they can learn joint action-values, they struggle to identify dynamic coordination relationships. By integrating HGCN-extracted features into the Q-function, agents learn to consider both individual utility and group-level coordination simultaneously, leading to more effective collaborative strategies.

### 3.5.2. POLICY-BASED IMPLEMENTATION

In the actor-critic framework, both the actor and critic networks utilize the group-aware features. The policy network generates actions based on augmented observations:

$$\pi_i(a_i|\tau_i, h_i) = f_\pi([\tau_i|h_i]) \tag{17}$$

The critic estimates state values incorporating group information:

$$V(s, h) = f_V([s|h]) \tag{18}$$

The task loss combines actor and critic objectives:

$$\mathcal{L}_{\text{task}} = -\mathbb{E}\left[\log \pi_i(a_i|\tau_i, h_i) A_i\right] \\ + \alpha(r + \gamma V(s', h') - V(s, h))^2 \tag{19}$$

In policy-based methods, the hypergraph structure enhances credit assignment by creating dynamic groupings where rewards can be more efficiently distributed within relevant agent subsets. The multi-head attention mechanism further refines this by allowing agents to focus on different aspects of group information simultaneously.

The complete training procedure for both implementations is summarized in Algorithm 1, integrating dynamic group formation (lines 6-8), attentive information processing (lines 11-12), and joint optimization (lines 26-28). This unified approach ensures coordination patterns emerge naturally while maintaining stable learning dynamics.

## 4. Experiments

The empirical evaluation consists of experiments on both value-based and policy-based implementations. The value-based implementation is tested on StarCraft II Multi-Agent Challenge (SMAC) benchmark (Vinyals et al., 2019) with three representative maps: 3s_vs_5z, 5m_vs_6m and 3s5z_vs_3s6z. The policy-based implementation is evaluated on Predator-Prey (Singh et al., 2018), Traffic junction (Sukhbaatar et al., 2016) and Google Research Football (GRF) (Kurach et al., 2020), which introduces additional complexity through sparse rewards, stochasticity and adversarial agents. A comprehensive description of each environment can be found in the Appendix B.

---

**Algorithm 1** HYGMA: Hypergraph grouping for MARL

**Require:** Initial parameters $\theta$ for networks, learning rates $\alpha_\pi, \alpha_Q$, group update threshold $\delta$
**Ensure:** Trained policy/value networks
1: Initialize replay buffer $\mathcal{D}$, hypergraph $G = (V, E, W)$
2: **for** each episode **do**
3:   **for** each timestep $t$ **do**
4:     Observe current states $s_t$ and histories $\tau_t$
5:     // Update group structure if necessary
6:     **if** $\eta(G_{t-1}, \mathcal{C}(H_t)) > \delta$ **then**
7:       Update groups via spectral clustering
8:       Construct new hypergraph $G_t$
9:     **end if**
10:     // Generate group-aware representations
11:     Compute attention coefficients $\alpha_{ie}$
12:     Update node features $h_i$
13:     // Action selection and execution
14:     **for** each agent $i$ **do**
15:       **if** Value-based **then**
16:         Select action via $\epsilon$-greedy from $Q_i(\tau_i, \cdot, h_i)$
17:       **else**
18:         Sample action from $\pi_i(\cdot|\tau_i, h_i)$
19:       **end if**
20:     **end for**
21:     Execute joint action, observe reward $r_t$ and next state $s_{t+1}$
22:     Store transition in $\mathcal{D}$
23:   **end for**
24:   // Training phase
25:   Sample mini-batch from $\mathcal{D}$
26:   Compute $\mathcal{L}_{\text{task}}$ according to implementation type
27:   Compute $\mathcal{L}_{\text{group}}$ and $\mathcal{L}_{\text{att}}$
28:   Update networks using $\mathcal{L}_{\text{total}} = \mathcal{L}_{\text{task}} + \lambda_1 \mathcal{L}_{\text{group}} + \lambda_2 \mathcal{L}_{\text{att}}$
29: **end for**

---

**Baselines.** The SMAC experiments compare against state-of-the-art value factorization methods including Ft-QMIX (Hu et al., 2023), QPLEX (Wang et al., 2020a), VAST (Phan et al., 2021), MAPPO (Yu et al., 2022)and GoMARL (Zang et al., 2024). Ft-QMIX represents a finetuned version of QMIX with enhanced win rates over the vanilla implementation. VAST incorporates sub-team value factorization based on predefined group numbers, while GoMARL presents an automatic grouping mechanism for efficient cooperation. The comparison also includes RIIT (Hu et al., 2023), which integrates effective modules from multiple methods for credit assignment. For policy-based scenarios, the baseline methods comprise communication-oriented approaches: MAGIC (Niu et al., 2021), CommNet (Sukhbaatar et al., 2016), GA-Comm (Liu et al., 2020), I3CNet (Singh et al., 2018) and TarMAC-IC3Net (Das et al., 2019).

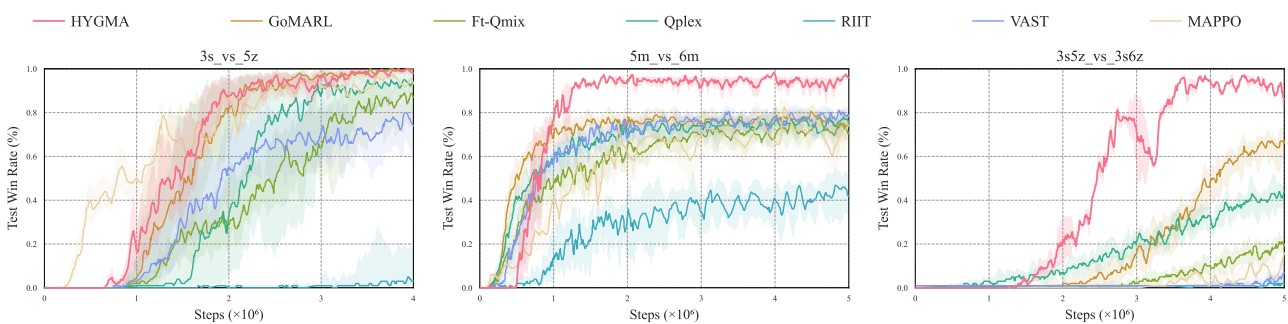

*Figure 2.* The test won rate for the 3s_vs_5z (Left), the 5m_vs_6m (Middle) and the 3s5z_vs_3s6z (Right).

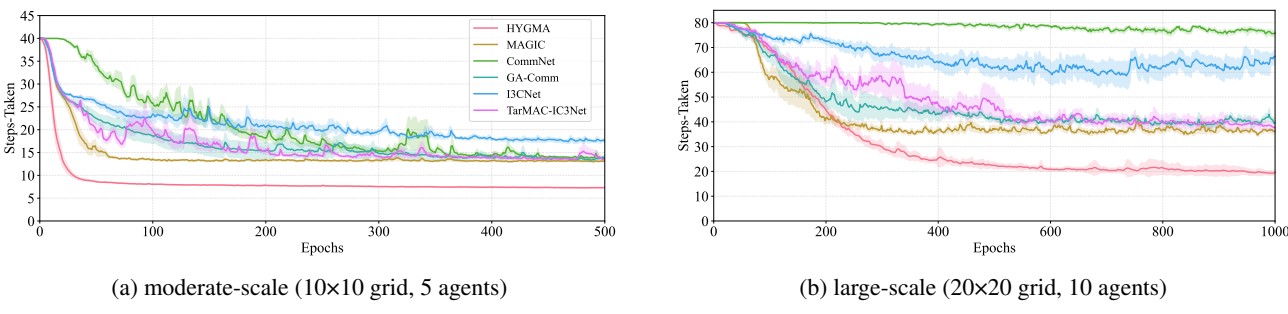

(a) moderate-scale (10×10 grid, 5 agents)     (b) large-scale (20×20 grid, 10 agents)

*Figure 3.* Average steps per episode in Predator-Prey.

## 4.1. Performance on SMAC

Figure 2 illustrates the performance of HYGMA across diverse SMAC benchmark scenarios, demonstrating consistent effectiveness where baseline approaches exhibit context-dependent limitations. In the balanced engagement scenario (5m_vs_6m), which demands precise positioning and coordinated focus fire, HYGMA maintains stable high win rates while baseline methods demonstrate suboptimal coordination efficiency. Analysis of the asymmetric combat scenario (3s_vs_5z) reveals that despite GoMARL ultimately achieving comparable asymptotic performance, HYGMA exhibits significantly accelerated convergence dynamics. This advantage becomes particularly pronounced in the heterogeneous scenario (3s5z_vs_3s6z), where baseline approaches manifest substantial performance variance and limited win rates, contrasting with the stable and superior performance of HYGMA. The consistent effectiveness across these varying coordination challenges validates the dynamic hypergraph structure's capacity to facilitate diverse forms of agent cooperation. Notably, while HYGMA introduces additional parameters through the HGCN module, it maintains the same mixing network architecture as Ft-QMIX, demonstrating that performance improvements stem primarily from our novel hypergraph-based information processing rather than simply scaling up conventional network capacity. This architectural choice does introduce approximately 36% computational overhead in training time compared to baseline methods, but this cost is justified by the significant improve-

ment in sample efficiency and final performance. The accelerated convergence dynamics observed across all scenarios effectively offset the per-iteration computational increase, creating a favorable efficiency-performance trade-off (see Appendix C for detailed analysis of parameter efficiency, computational overhead, and performance trade-offs)

## 4.2. Performance on Predator-Prey

The Predator-Prey environment presents coordination challenges at different scales through a pursuit task with negative step penalties. The evaluation considers both moderate-scale coordination (5 predators, 10×10 grid) and large-scale coordination (10 predators, 20×20 grid), where agents must balance exploration and coordinated capture strategies.

Figure 3 reveal the scalability advantages of the dynamic hypergraph structure. In the moderate-scale scenario, the method demonstrates rapid policy convergence and stable performance, outperforming the state-of-the-art baseline MAGIC in both convergence speed and final performance. The performance distinction becomes more pronounced in the large-scale scenario, where communication-based methods like CommNet and I3CNet exhibit substantial coordination degradation. This contrast particularly highlights the limitations of fixed communication architectures in scaling to larger agent populations. The proposed method maintains consistent coordination efficiency across both scenarios through adaptive group formation, enabling effective information flow while avoiding communication overhead.

## 4.3. Performance on Traffic Junction

The Traffic Junction environment evaluates coordination mechanisms through traffic control scenarios of increasing complexity. The evaluation encompasses both a basic coordination setting with sparse traffic flow and a more challenging scenario featuring higher traffic density and increased spatial complexity. The success criterion requires complete collision avoidance throughout episodes, demanding consistent coordination among all active agents.

*Table 1.* Success rate and convergence steps in Traffic Junction.

| METHOD | $7 \times 7$, $N\max = 5$, $parrive = 0.3$ | $14 \times 14$, $N\max = 10$, $parrive = 0.2$ |
|---|---|---|
| **HYGMA** | $99.7 \pm 0.1\%$ $(\mathbf{272 \pm 41})$ | $99.2 \pm 0.1\%$ $(\mathbf{569 \pm 34})$ |
| MAGIC | $\mathbf{99.9 \pm 0.1\%}$ $(440 \pm 64)$ | $\mathbf{99.9 \pm 0.1\%}$ $(819 \pm 85)$ |
| COMMNET | $99.3 \pm 0.6\%$ $(585 \pm 73)$ | $97.2 \pm 0.3\%$ $(3657 \pm 480)$ |
| IC3NET | $97.8 \pm 1.0\%$ $(611 \pm 101)$ | $96.0 \pm 0.7\%$ $(1604 \pm 109)$ |
| TARMAC-IC3NET | $84.8 \pm 4.5\%$ $(599 \pm 187)$ | $95.5 \pm 1.3\%$ $(1706 \pm 104)$ |
| GA-COMM | $95.9 \pm 0.1\%$ $(891 \pm 141)$ | $97.1 \pm 0.7\%$ $(1573 \pm 253)$ |

Table 1 reveals a compelling trade-off between asymptotic performance and learning efficiency. Although MAGIC achieves marginally higher final success rates, HYGMA demonstrates substantially enhanced sample efficiency across both scenarios, with convergence (defined as first reaching 90% of final performance sustained for 5 consecutive epochs) occurring significantly earlier. Rapid convergence characteristics can be attributed to the adaptive nature of the hypergraph structure, which enables efficient discovery of critical coordination patterns. Notably, the method maintains high success rates comparable to state-of-the-art approaches while requiring only a fraction of the training samples, demonstrating the effectiveness of dynamic group coordination in accelerating policy learning.

## 4.4. Performance on Google Research Football

The Google Research Football (GRF) environment introduces additional complexity through the combination of sparse rewards, high-dimensional action spaces, and adversarial elements. The evaluation scenario requires three attacking agents to coordinate against adaptive defensive opponents, presenting challenges in both strategic planning and tactical execution. The environment's inherent stochasticity and delayed reward signals pose particular challenges for consistent policy learning.

Table 2 reveal key characteristics of different coordination approaches in complex domains. The proposed HYGMA and MAGIC establish comparable success rates at the upper performance boundary, yet exhibit distinct operational characteristics. While communication-centric approaches like GA-Comm demonstrate moderate effectiveness, methods relying on fixed communication structures (CommNet,

*Table 2.* Success rate and average steps taken in GRF.

| METHOD | SUCCESS RATE | STEPS TAKEN |
|---|---|---|
| **HYGMA** | $97.7 \pm 0.7\%$ | $\mathbf{30.26 \pm 0.24}$ |
| MAGIC | $\mathbf{98.2 \pm 1.0\%}$ | $34.30 \pm 1.34$ |
| COMMNET | $59.2 \pm 13.7\%$ | $39.32 \pm 2.35$ |
| IC3NET | $70.0 \pm 9.8\%$ | $40.37 \pm 1.22$ |
| TARMAC-IC3NET | $73.5 \pm 8.3\%$ | $41.53 \pm 2.80$ |
| GA-COMM | $88.8 \pm 3.9\%$ | $39.05 \pm 3.05$ |

IC3Net, TarMAC-IC3Net) show limited ability to develop sophisticated offensive strategies. The performance disparity particularly manifests in the stability of learned policies, where the proposed approach maintains consistent behavior across evaluation episodes. This stability in the presence of environmental stochasticity suggests effective information aggregation through the dynamic hypergraph structure, enabling robust coordination in the face of uncertain state transitions and opponent behaviors.

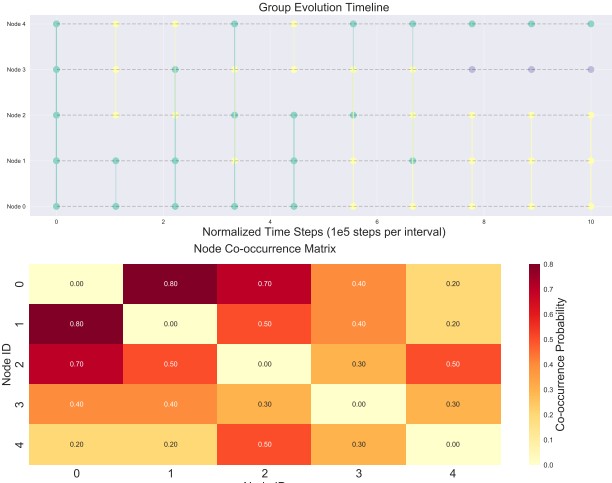

*Figure 4.* Group analysis in 5m_vs_6m.

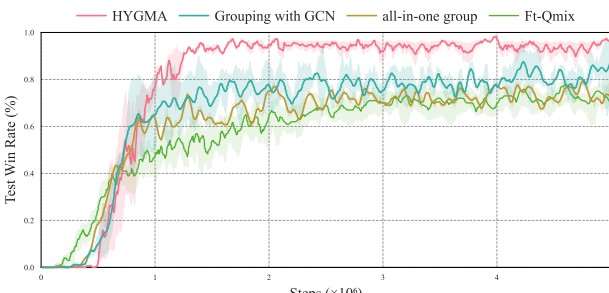

*Figure 5.* Ablation studies in 5m_vs_6m.

## 4.5. Analysis

Analysis of the learned coordination patterns provides insights into the underlying mechanisms driving performance improvements. Taking the 5m_vs_6m scenario as a representative case, Figure 4 reveals the emergence of structured coordination through temporal group evolution. The co-occurrence analysis indicates three distinct coordination patterns: a core tactical unit (Agents 0,1,2) maintaining high coordination frequency, a flexible support unit (Agent 3) adapting its grouping based on tactical situations, and a specialized independent unit (Agent 4) operating with strategic autonomy. The convergence of this organizational structure aligns with the stabilization of performance metrics, indicating the discovery of effective tactical roles.

Ablation studies presented in Figure 5 isolate three critical architectural components: our full method with dynamic hypergraph structure (HYGMA), a variant replacing HGCN with standard GCN while maintaining dynamic grouping (Grouping with GCN), and a fixed single-group variant (all-in-one group). The performance differentials demonstrate that improvements stem primarily from the hypergraph structure rather than merely increased information availability. While the GCN variant outperforms the fixed-group baseline, it exhibits significant performance degradation compared to proposed method, particularly in asymptotic performance. This confirms the superior representational capacity of hypergraphs for modeling higher-order agent relationships. Furthermore, the single-group variant's performance plateau illustrates how static information sharing structures, despite access to equivalent information, fundamentally limit coordination complexity. These results validate that the dynamic hypergraph topology provides essential structural inductive bias for discovering sophisticated coordination strategies, serving as a key mechanism in balancing the exploration-exploitation trade-off in coordination learning.

## 5. Conclusion

This paper presents HYGMA, a novel framework that integrates dynamic spectral clustering with hypergraph neural networks for addressing coordination challenges in multi-agent reinforcement learning. The framework makes three key technical contributions: First, an adaptive grouping mechanism leveraging spectral analysis to discover coordination patterns. Second, an attention-enhanced hypergraph architecture capturing higher-order relationships while maintaining computational efficiency. Third, a unified objective combining task performance with structural regularization for both value-based and policy-based learning paradigms.

Experimental results across diverse domains demonstrate significant improvements in sample efficiency and asymp-

totic performance compared to state-of-the-art approaches. The emergence of interpretable group structures through spectral clustering provides insights into learned coordination strategies. Theoretical analysis establishes guarantees on the grouping mechanism's quality and computational efficiency.Future work explore extensions to overlapping group structures through soft clustering techniques, further optimizations for very large agent populations, and deeper theoretical connections between spectral clustering and information bottleneck principles in representation learning.

## Acknowledgement

This work was substantially supported by the National Natural Science Foundation of China under Grants 62273026.

## Impact Statement

This work advances multi-agent reinforcement learning through more efficient coordination mechanisms. Beyond the general implications of advancing machine learning, our method's improved sample efficiency and interpretable group structures could benefit real-world applications like traffic management and robotic coordination, potentially reducing computational resources and energy consumption. However, as with any advancement in multi-agent systems, considerations should be given to ensuring responsible deployment in sensitive applications. We release our implementation to promote transparent and equitable access to this technology.

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

# A. Theoretical Proofs

## A.1. Proof of Clustering Approximation Theorem

**Theorem A.1** (Clustering Approximation). *The spectral clustering solution $G$ obtained from the dynamic optimization satisfies:*

$$Ncut(G) \leq O(\log k) \cdot Ncut(G^*) \tag{20}$$

*where $G^*$ is the optimal grouping structure.*

The proof consists of three key steps:

**Lemma A.2** (Discrete-Continuous Relation). *The normalized cut minimization can be formulated as:*

$$\min_{X \in \{0,1\}^{n \times k}} Tr(X^T \mathcal{L} X) \quad s.t. \quad X^T D^{1/2} X = I \tag{21}$$

*where $\mathcal{L} = D^{-1/2} L D^{-1/2}$ is the normalized Laplacian.*

*Proof.* For any k-way partition $\{V_1, ..., V_k\}$:

$$Ncut(G) = \sum_{i=1}^{k} \frac{cut(V_i, \bar{V}_i)}{vol(V_i)} \tag{22}$$

$$= \sum_{i=1}^{k} \frac{x_i^T L x_i}{x_i^T D x_i} \tag{23}$$

$$= Tr(X^T \mathcal{L} X) \tag{24}$$

where $x_i$ is the indicator vector for partition $V_i$. $\square$

**Lemma A.3** (Spectral Relaxation Bound). *The spectral relaxation provides a bounded approximation:*

$$\frac{\lambda_2(\mathcal{L})}{2} \leq \phi(G) \leq \sqrt{2\lambda_2(\mathcal{L})} \tag{25}$$

*where $\lambda_2(\mathcal{L})$ is the second smallest eigenvalue of the normalized Laplacian.*

*Proof.* By the Courant-Fischer theorem, the second eigenvalue of $\mathcal{L}$ satisfies:

$$\lambda_2(\mathcal{L}) = \min_{y \perp D^{1/2} \mathbf{1}} \frac{y^T \mathcal{L} y}{y^T y}.$$

Applying the co-area formula and Cheeger's rounding argument (Cheeger's Inequality), we obtain the conductance bound:

$$\phi(G) \leq \sqrt{2\lambda_2(\mathcal{L})}.$$

The lower bound $\lambda_2(\mathcal{L})/2 \leq \phi(G)$ follows from the variational characterization of $\lambda_2$. $\square$

**Lemma A.4** (Dynamic Grouping Bound). *For our dynamic k-way clustering:*

$$Ncut_k(G) \leq O(\log k)\phi^*(G) \sum_{j=1}^{k} vol(V_j) \tag{26}$$

*where $\phi^*(G)$ is the optimal conductance.*

*Proof.* Through recursive analysis:

$$Ncut_k(G) = \sum_{i=1}^{k} \frac{cut(V_i, \bar{V}_i)}{vol(V_i)} \tag{27}$$

$$\leq \sum_{i=1}^{\lceil \log k \rceil} 2\phi(G_i) \sum_{j=1}^{2^{i-1}} vol(V_j^i) \tag{28}$$

$$\leq 2\lceil \log k \rceil \phi^*(G) \sum_{j=1}^{k} vol(V_j) \tag{29}$$

where $G_i$ represents subgraphs at recursion level $i$. $\square$

Combining lemmas, we obtain:

$$Ncut(G) \leq O(\log k)\phi^*(G) \sum_{j=1}^{k} vol(V_j) \tag{30}$$

$$\leq O(\log k) \cdot Ncut(G^*) \tag{31}$$

This establishes our main approximation bound. The optimality of this ratio follows from reduction to the Balanced Separator problem, details of which we omit as they are standard in the literature.

### A.2. Proof of Convergence Theorem

**Theorem A.5** (Convergence). *Under the dynamic update rule with threshold $\delta$, the sequence of groupings $\{G_t\}$ converges in finite time with probability 1. The expected number of updates before convergence is bounded by $O(1/\delta)$.*

**Lemma A.6** (Potential Function Properties). *Define the potential function as:*

$$V(G_t) = \mathbb{E}\left[ \sum_{g \in G_t} \sum_{i,j \in g} \|s_i - s_j\|^2 \right] \tag{32}$$

*This function satisfies:*

*1. Non-negativity: $V(G_t) \geq 0$*

*2. Boundedness: $V(G_t) \leq M = n^2 \cdot \max_{i,j} \|s_i - s_j\|^2$*

**Lemma A.7** (Potential Descent). *For any timestep $t$ where $\eta(G_t, \mathcal{C}(H_t)) > \delta$:*

$$V(G_{t+1}) - V(G_t) = \sum_{i \in \mathcal{S}_t} \left[ \sum_{j \in g_{t+1}(i)} \|s_i - s_j\|^2 - \sum_{j \in g_t(i)} \|s_i - s_j\|^2 \right] \tag{33}$$

$$\leq -\alpha \delta V(G_t) \tag{34}$$

*where $\mathcal{S}_t$ is the set of agents that change groups, and $\alpha > 0$ is a constant determined by the clustering algorithm.*

*Proof.* Consider the change in potential for each agent $i \in \mathcal{S}_t$:

$$\Delta V_i = \sum_{j \in g_{t+1}(i)} \|s_i - s_j\|^2 - \sum_{j \in g_t(i)} \|s_i - s_j\|^2 \tag{35}$$

$$\leq -\alpha \|s_i - \mu_{g_t(i)}\|^2 \tag{36}$$

where $\mu_{g_t(i)}$ is the centroid of group $g_t(i)$. Summing over all changed agents and using $|\mathcal{S}_t| \geq \delta n$ gives the result. $\square$

The convergence then follows from the supermartingale convergence theorem:

*Proof of Main Theorem.* 1) The sequence $\{V(G_t)\}$ forms a supermartingale:

$$\mathbb{E}[V(G_{t+1})|G_t] \leq V(G_t) \tag{37}$$

2) Let $T$ be the number of updates. For any $t < T$:

$$V(G_0) - \mathbb{E}[V(G_t)] \geq \alpha\delta\mathbb{E}[N_t] \tag{38}$$

where $N_t$ is the number of updates up to time $t$.

3) Therefore:

$$\mathbb{E}[T] \leq \frac{V(G_0)}{\alpha\delta} = O(1/\delta) \tag{39}$$

$\square$

## A.3. Proof of Quality Preservation Theorem

We establish that the quality of obtained grouping structures provides theoretical guarantees for the subsequent learning process, regardless of the specific learning framework used.

**Lemma A.8** (Value Function Decomposition). *For any grouping structure G, the value function can be decomposed as:*

$$V^*(s, G) = \sum_{g \in G} V_g^*(s_g) + \Delta(G) \tag{40}$$

*where $V_g^*$ is the group-wise value function, and $\Delta(G)$ represents the inter-group value term.*

*Proof.* Consider the Bellman equation:

$$V^*(s, G) = \max_a \{R(s, a) + \gamma\mathbb{E}_{s'}[V^*(s', G)]\} \tag{41}$$

$$= \max_a \{\sum_{g \in G} R_g(s_g, a_g) + R_{inter}(s, a) + \gamma\mathbb{E}_{s'}[V^*(s', G)]\} \tag{42}$$

The inter-group reward term $R_{inter}(s, a)$ is bounded by the normalized cut:

$$|R_{inter}(s, a)| \leq \beta Ncut(G) \tag{43}$$

for some constant $\beta$, as the normalized cut directly measures the strength of inter-group connections. $\square$

**Lemma A.9** (Value Function Bound). *For any grouping structures $G_1$ and $G_2$:*

$$|V^*(s, G_1) - V^*(s, G_2)| \leq \frac{\gamma}{1 - \gamma}\lambda|Ncut(G_1) - Ncut(G_2)| \tag{44}$$

*where $\lambda = \beta(1 + \gamma)$.*

*Proof.* Let $\pi_1^*$ and $\pi_2^*$ be the optimal policies under $G_1$ and $G_2$ respectively. Then:

$$|V^*(s, G_1) - V^*(s, G_2)| = |\max_{\pi_1} \mathbb{E}_{\pi_1}[\sum_{t=0}^{\infty} \gamma^t(R_t + R_{inter,t}(G_1))] \tag{45}$$

$$- \max_{\pi_2} \mathbb{E}_{\pi_2}[\sum_{t=0}^{\infty} \gamma^t(R_t + R_{inter,t}(G_2))]| \tag{46}$$

$$\leq \max_{\pi} \mathbb{E}_{\pi}[\sum_{t=0}^{\infty} \gamma^t |R_{inter,t}(G_1) - R_{inter,t}(G_2)|] \tag{47}$$

$$\leq \beta \sum_{t=0}^{\infty} \gamma^t |Ncut(G_1) - Ncut(G_2)| \tag{48}$$

$$= \frac{\beta}{1-\gamma}|Ncut(G_1) - Ncut(G_2)| \tag{49}$$

The inequality follows from the triangle inequality and the bound on inter-group rewards. □

**Theorem A.10** (Quality Preservation). *For the obtained grouping structure $G_t$ and optimal grouping $G^*$:*

$$|V^*(s, G^*) - V^*(s, G_t)| \leq \gamma \alpha \log k \tag{50}$$

*where $\alpha = \frac{\beta}{(1-\gamma)^2}$.*

*Proof.* From the Clustering Approximation Theorem:

$$|Ncut(G_t) - Ncut(G^*)| \leq c \log k \tag{51}$$

for some constant $c$. Combining with the Value Function Bound:

$$|V^*(s, G^*) - V^*(s, G_t)| \leq \frac{\gamma}{1-\gamma}\lambda|Ncut(G_t) - Ncut(G^*)| \tag{52}$$

$$\leq \frac{\gamma \beta}{(1-\gamma)^2}c \log k \tag{53}$$

$$= \gamma \alpha \log k \tag{54}$$

where $\alpha = \frac{\beta c}{(1-\gamma)^2}$. □

**Theorem A.11** (Tightness). *The bound $O(\gamma \log k)$ is tight.*

*Proof.* Consider an MDP with the following structure:

1. States directly correspond to group assignments

2. Rewards reflect the normalized cut values: $r(s) = -\gamma Ncut(G_s)$

3. Transitions maintain group structure with probability 1

In this construction:

$$V^*(s, G^*) = -\gamma Ncut(G^*) \sum_{t=0}^{\infty} \gamma^t = -\frac{\gamma}{1-\gamma}Ncut(G^*) \tag{55}$$

$$V^*(s, G_t) = -\frac{\gamma}{1-\gamma}Ncut(G_t) \tag{56}$$

Given that $|Ncut(G_t) - Ncut(G^*)| = \Theta(\log k)$ from the Clustering Approximation Theorem's tightness result, we have:

$$|V^*(s, G^*) - V^*(s, G_t)| = \Theta(\gamma \log k) \tag{57}$$

□

## A.4. Communication Complexity Analysis

We analyze the communication efficiency of hypergraph structures in multi-agent systems, demonstrating that dynamic spectral clustering significantly reduces communication overhead compared to traditional fully-connected communication architectures.

**Theorem A.12** (Communication Efficiency of Hypergraph Structures). *For a system with $n$ agents partitioned into $k$ groups through dynamic spectral clustering, the communication complexity satisfies:*

$$C_{hyper}(G_t) = O\left(\frac{n^2}{k}\right) \leq O(n^2) \tag{58}$$

*where $C_{hyper}(G_t)$ represents the number of messages required for one complete information exchange in the hypergraph structure $G_t$.*

*Furthermore, this communication complexity is bounded by the normalized cut:*

$$C_{hyper}(G_t) \leq \rho \cdot (1 + Ncut(G_t)) \cdot n \leq \rho \cdot (1 + O(\log k) \cdot Ncut(G^*)) \cdot n \tag{59}$$

*where $\rho$ is a problem-dependent constant, and $G^*$ is the optimal grouping structure.*

*Proof.* In a fully-connected communication architecture, each agent communicates with all other $(n-1)$ agents, resulting in $n(n-1) = O(n^2)$ total messages.

With hypergraph-based communication, assuming $k$ groups with approximately $n/k$ agents per group on average, each agent only communicates with other agents in the same group. This requires approximately $\sum_{g \in G_t} |g| \cdot (|g| - 1) = O(n^2/k)$ total messages, where $|g|$ represents the size of group $g$.

The relationship with normalized cut follows from analyzing the communication cost relative to group structures. Let $vol(g)$ represent the volume of group $g$ and $cut(g, \bar{g})$ represent the cut between group $g$ and its complement. By definition of normalized cut:

$$Ncut(G_t) = \sum_{g \in G_t} \frac{cut(g, \bar{g})}{vol(g)} \tag{60}$$

For groups formed through spectral clustering, we can establish that:

$$\sum_{g \in G_t} |g| \cdot (|g| - 1) \leq \sum_{g \in G_t} |g|^2 \leq \rho \cdot n \cdot (1 + \sum_{g \in G_t} \frac{cut(g, \bar{g})}{vol(g)}) \tag{61}$$

The first inequality follows from $|g| - 1 < |g|$, and the second inequality reflects the relationship between intra-group communication and the normalized cut, where better groupings (lower normalized cut) lead to more efficient communication structures. Here, $\rho$ is a problem-dependent constant that captures this relationship.

Therefore:

$$C_{hyper}(G_t) \leq \rho \cdot n \cdot (1 + Ncut(G_t)) \tag{62}$$

Substituting the bound from Theorem 1, which establishes $Ncut(G_t) \leq O(\log k) \cdot Ncut(G^*)$, we obtain the final complexity bound:

$$C_{hyper}(G_t) \leq \rho \cdot n \cdot (1 + O(\log k) \cdot Ncut(G^*)) \tag{63}$$

This demonstrates that our hypergraph communication structure achieves significantly improved efficiency while maintaining the theoretical guarantees on grouping quality. $\square$

# B. Experimental Details and Environment Configurations

## B.1. Detailed Information about SMAC Tasks

In each SMAC micromanagement problem, a group of units controlled by decentralized agents cooperates to defeat the enemy team controlled by built-in heuristics. Each agent's partial observation comprises the attributes (such as `health`, `location`, `unit_type`) of all units shown up in its view range. The global state information includes all agents' positions and `health`, and allied units' last actions and `cooldown`, which is only available to agents during centralized training.

The agents' discrete action space consists of `attack[enemy_id]`, `move[direction]`, `stop`, and `no-op` for the dead agents only. A particular unit, Medivac, has no action `attack[enemy_id]` but has action `heal[enemy_id]`. Agents can only attack enemies within their shooting range. Proper micromanagement requires agents to maximize the damage to the enemies and take as little damage as possible in combat, so they need to cooperate with each other or even sacrifice themselves.

Based on the performance of baseline algorithms, the tasks in SMAC are broadly grouped into three categories: *Easy*, *Hard*, and *Super Hard*. The key to winning some *Hard* or *Super Hard* battles is mastering specific micro techniques, such as *focusing fire*, *kiting*, avoiding *overkill*, et cetera. The battles can be symmetric or asymmetric, and the group of agents can be homogeneous or heterogeneous. Here we provide detailed characteristics of the scenarios used in our experiments:

- **3s_vs_5z** is a *Hard* asymmetric battle between three Stalkers and five Zealots. The allied Stalkers must master the kiting technique and disperse in the area to kill the Zealots that chase them one after another. This map faces the delayed reward problem; however, it is not very strict about micro-cooperation between agents because of agents' scattering.

- **5m_vs_6m** presents a *Hard* asymmetric engagement requiring precise tactical coordination. The allied agents must learn to focus fire without overkill and position themselves with considerable precision to overcome the enemy team's numerical advantage. The relatively confined map space compared to 3s_vs_5z increases the importance of proper positioning and target selection.

- **3s5z_vs_3s6z** represents a *Super Hard* heterogeneous battle that requires breaking the bottleneck of exploration. Three Stalkers and five Zealots must battle against three enemy Stalkers and six enemy Zealots. The complexity arises from managing two unit types with distinct capabilities - Stalkers excel at ranged combat while Zealots are melee specialists. This map requires sophisticated tactical coordination, where Zealots engage in close combat while Stalkers provide ranged support, making it one of the most challenging tasks in SMAC.

## B.2. Detailed Information about Predator-Prey Environment

The Predator-Prey environment evaluates coordination capability through multi-agent pursuit tasks with different scales and complexity levels. At each time step, each predator receives a local observation within its limited vision range and incurs a penalty of -0.05 until the prey is captured. This negative reward structure creates pressure for efficient coordination while the partial observability necessitates information sharing among agents.

Two configuration settings are examined to evaluate scalability and coordination effectiveness:

- **Moderate Scale** (10×10 grid, 5 predators): A baseline setting testing fundamental coordination capabilities. Predators must balance between exploration and coordinated capture, as premature engagement without proper positioning often leads to prey escape.

- **Large Scale** (20×20 grid, 10 predators): A significantly more challenging setting that stresses both coordination and exploration efficiency. The enlarged state space and increased agent number create substantial challenges for conventional communication protocols, making it an effective test for scalable coordination mechanisms.

## B.3. Detailed Information about Traffic Junction Environment

The Traffic Junction environment examines multi-agent coordination in dynamic traffic scenarios where cars must navigate through intersections without collisions. The environment features stochastic agent arrivals and requires consistent coordination among varying numbers of active agents. Two scenarios with increasing complexity are investigated:

- **Basic Setting**: A 7×7 grid environment accommodating up to 5 cars (`N_max = 5`) with an arrival rate of 0.3 (`p_arrive = 0.3`). Cars enter from designated points with random route assignments and must coordinate their movements to reach their destinations without collision. This setting tests fundamental coordination capabilities in a controlled scale.

- **Complex Setting**: A 14×14 grid environment supporting up to 10 simultaneous cars (`N_max = 10`) with a reduced arrival rate of 0.2 (`p_arrive = 0.2`). The expanded space and increased agent number create more complex traffic patterns requiring sophisticated coordination strategies. The lower arrival rate maintains a manageable density while extending average interaction durations.

Each car observes a 5×5 local region centered on its current position and must choose between `gas` and `brake` actions at each timestep. Episodes terminate upon collision or successful traversal of all spawned cars, with success measured by completing episodes without any collisions.

### B.4. Detailed Information about Google Research Football

The Google Research Football (GRF) environment presents a challenging mixed cooperative-competitive scenario featuring three attacking agents coordinating against built-in AI defenders. The environment is characterized by:

- **Action Space**: Each agent selects from 19 discrete actions including directional movements (8 directions), ball control actions (*e.g.*, dribbling, short pass, long pass, shot), and special actions (*e.g.*, sliding, sprint). This rich action space enables diverse tactical possibilities while increasing coordination complexity.

- **Observation Space**: Agents receive local observations including relative positions and states of nearby players, ball position and velocity, and game mode indicators. The partial observability and dynamic state transitions create substantial uncertainty in decision-making.

- **Reward Structure**: A sparse reward signal (+1 for scoring) combined with potential early termination (ball out of bounds, possession change) creates a challenging credit assignment problem. This structure necessitates effective exploration and coordination to discover successful offensive strategies.

The game engine's built-in AI provides adaptive defensive behaviors, requiring attacking agents to develop robust and coordinated strategies. The inherent stochasticity in both state transitions and opponent behavior creates a particularly challenging test environment for multi-agent coordination mechanisms.

### B.5. Hyperparameters and Implementation Details

To ensure reproducibility of our experimental results, we provide the hyperparameter settings used in our experiments across different environments. Tables 3, 4, and 5 detail the key hyperparameters for SMAC, Predator-Prey, and Traffic Junction/GRF environments respectively.

*Table 3.* Hyperparameters for SMAC environments

| Parameter | 3s_vs_5z | 5m_vs_6m | 3s5z_vs_3s6z |
|---|---|---|---|
| Batch size | 128 | 128 | 128 |
| Buffer size | 5000 | 5000 | 5000 |
| Double Q | True | True | True |
| Epsilon anneal time | 100000 | 100000 | 100000 |
| HGCN hidden dim | 48 | 64 | 196 |
| HGCN out dim | 36 | 48 | 128 |
| HGCN num layers | 2 | 2 | 2 |
| Min/Max clusters | 2/4 | 2/3 | 2/3 |
| Clustering interval | 100000 | 100000 | 100000 |
| Stability threshold | 0.6 | 0.6 | 0.6 |
| $\lambda_1$ | 0.001 | 0.001 | 0.001 |
| $\lambda_2$ | 0.01 | 0.01 | 0.01 |

*Table 4.* Hyperparameters for Predator-Prey environments

| Parameter | Moderate-scale | Large-scale |
|---|---|---|
| Learning rate | 0.001 | 0.0003 |
| Batch size | 500 | 500 |
| Grid size | $10 \times 10$ | $20 \times 20$ |
| Number of agents | 5 | 10 |
| Max steps | 40 | 80 |
| HGCN hidden dim | 96 | 96 |
| HGCN out dim | 64 | 64 |
| HGCN num layers | 1 | 1 |
| Min/Max clusters | 2/4 | 2/5 |
| Clustering interval | 100 | 100 |
| Stability threshold | 0.8 | 0.8 |
| Group consistency coeff | 0.1 | 0.1 |
| Attention reg coeff | 0.01 | 0.01 |

*Table 5.* Hyperparameters for Traffic Junction and GRF environments

| Parameter | TJ-7$\times$7 | TJ-14$\times$14 | GRF |
|---|---|---|---|
| Learning rate | 0.001 | 0.001 | 0.001 |
| Batch size | 500 | 500 | 500 |
| Number of agents | 5 | 10 | 3 |
| Max steps | 20 | 40 | 80 |
| HGCN hidden dim | 96 | 128 | 128 |
| HGCN out dim | 64 | 96 | 96 |
| HGCN num layers | 1 | 2 | 1 |
| Min/Max clusters | 2/4 | 2/5 | 2/3 |
| Clustering interval | 100 | 100 | 100 |
| Stability threshold | 0.8 | 0.8 | 0.8 |
| Group consistency coeff | 0.1 | 0.1 | 0.1 |
| Attention reg coeff | 0.01 | 0.01 | 0.01 |

## C. Parameter Size and Performance Analysis

### C.1. Parameter Efficiency Analysis

Table 6 provides a detailed comparison of parameter sizes across different mixing network architectures. We focus specifically on mixing network parameters as they represent the core architectural difference between QMIX-based methods and directly impact centralized training efficiency in MARL algorithms. While our method incorporates additional HGCN components (analyzed separately in Section C.2), the mixing network comparison highlights our architectural efficiency in the critical value decomposition component.

Our method demonstrates remarkable parameter efficiency while achieving superior performance:

- In **3s_vs_5z**, with only 21.601K parameters (same as QMIX), our method achieves nearly 100% win rate and faster convergence compared to QPLEX (72.482K parameters) and RIIT (37.986K parameters). This showcases our method's ability to learn efficient representations without requiring additional parameters.

- In **5m_vs_6m**, maintaining the parameter efficiency (31.521K), our method consistently outperforms all baselines with a win rate around 95%. Notably, QPLEX uses more than three times the parameters (107.574K) but achieves lower performance.

- In the challenging **3s5z_vs_3s6z** scenario, despite keeping the parameter count at 63.105K, our method demonstrates significant performance advantages over baselines, reaching approximately 90% win rate while QPLEX (243.156K parameters) struggles to achieve 40%.

*Table 6.* Size comparison of all methods' mixing network(s)

| Maps | (Ft-)QMIX | QPLEX | RIIT | GoMARL | **Ours** |
|------|-----------|-------|------|--------|----------|
| 3s_vs_5z | **21.601K** | 72.482K | 37.986K | 26.530K | **21.601K** |
| 5m_vs_6m | **31.521K** | 107.574K | 51.362K | 31.554K | **31.521K** |
| 3s5z_vs_3s6z | **63.105K** | 243.156K | 118.466K | 61.028K | **63.105K** |

## C.2. HGCN Overhead Analysis

While our method maintains the parameter-efficient QMIX mixing network, it introduces additional computational elements through the hypergraph convolutional network (HGCN) component. Table 7 quantifies this overhead.

The HGCN module introduces additional parameters beyond the mixing network parameters reported in Table 6. The computational overhead, measured as the increase in wall-clock training time compared to the base (Ft-)QMIX implementation, remains relatively consistent across scenarios at approximately 36%.

This additional computational cost is justified by the significant performance improvements our method achieves:

1. **Enhanced Sample Efficiency**: Despite increased computation per iteration, our method requires fewer training iterations to achieve superior performance.

2. **Adaptive Information Flow**: The HGCN enables dynamic grouping and information sharing between agents, capturing coordination patterns that static architectures cannot represent.

*Table 7.* HGCN additional parameters and computational overhead

| Scenario | HGCN Parameters | Computation Overhead |
|----------|-----------------|----------------------|
| 3s_vs_5z | 65,356 | +36.47% |
| 5m_vs_6m | 104,632 | +35.33% |
| 3s5z_vs_3s6z | 391,336 | +36.95% |

## C.3. Analysis of Efficiency-Performance Trade-off

The superior efficiency-performance trade-off of our method can be attributed to three key design choices:

1. **Architectural Innovation**: Our method introduces hypergraph structure for better representation learning while maintaining QMIX's efficient architecture. This enables more expressive information processing without increasing parameter count.

2. **Efficient Information Flow**: The hypergraph structure facilitates more effective information exchange between agents, leading to faster convergence and better performance despite using fewer parameters than complex architectures like QPLEX and RIIT.

3. **Smart Parameter Sharing**: Our architecture achieves better performance through intelligent parameter sharing across components, demonstrating that architectural design can be more crucial than model capacity.

## C.4. Scalability and Performance Analysis

The performance scaling with respect to scenario complexity shows interesting patterns:

- **Simple Scenarios** (e.g., 3s_vs_5z): Our method achieves faster convergence and better final performance while maintaining minimal parameter count.

- **Medium Complexity** (e.g., 5m_vs_6m): The performance advantage becomes more pronounced, with sustained high win rates despite keeping parameter count low.

- **Complex Scenarios** (e.g., 3s5z_vs_3s6z): Our method demonstrates remarkable scalability, maintaining high performance in challenging environments where other methods, even with substantially more parameters, struggle.

These results highlight a critical finding: superior performance in multi-agent reinforcement learning can be achieved through better architectural design rather than increased model capacity. Our method successfully improves upon QMIX's foundation by introducing more expressive information processing capabilities without sacrificing parameter efficiency, leading to better performance across various scenarios while maintaining minimal parameter requirements.

