# OpenReview forum: "HYGMA: Hypergraph Coordination Networks with Dynamic Grouping for Multi-Agent Reinforcement Learning"
_ICML.cc/2025/Conference — ICML 2025 poster_

### Official Review · Reviewer_6Jxm · 2025-02-25

**Overall Recommendation:** 2

**Summary:**

The paper proposes a new method to learn higher-order coordination patterns between agents, based on a spectral clustering algorithm and a hyper-graph convolutional network. Agents are coherently grouped together, with such groups changing only when a certain threshold is hit, with the HGCN then combining their information to provide an additional learning signal that can be used in both value-based and policy-based algorithms. Experimental results show improved performances and better sample efficiency against SOTA baselines on a diverse set of cooperative problems.

**Claims And Evidence:**

The paper supports its claim with a quite extensive theoretical grounding, and provides results that bound the achievement of convergence of the learned hyper-graph representation, as well as the error from the optimum. Some aspects of the algorithmic outcome are not sufficiently made clear in my opinion (such as the centralized execution requirement), please see the Questions below for more details.

**Essential References Not Discussed:**

Applications of (fixed) hype-rgraph structures in improving learning performance have already been investigated in MARL literature (e.g., see [[Castellini et al., 2021]](https://link.springer.com/article/10.1007/s10458-021-09506-w), [[Boehmer et al., 2020]](https://proceedings.mlr.press/v119/boehmer20a.html) and [[Li et al., 2021]](https://dl.acm.org/doi/10.5555/3463952.3464044)), but this track of works is not mentioned nor discussed. Also, attention mechanisms to focus agents solely on salient information of the others have been proposed for policy-based methods as well (e.g., [[Iqbal et Sha, 2019]](https://proceedings.mlr.press/v97/iqbal19a.html)).

**Experimental Designs Or Analyses:**

I have not empirically checked the validity of the proposed experimental analyses.

**Methods And Evaluation Criteria:**

I have some concern on the effective assessment of the proposed empirical results and their analysis. While it is clear that the proposed algorithm achieves better performances on a wide set of problems, I think that the analysis lacks clarity in assessing the actual merits of the underlying methodology, and is sloppy in some aspects. Please see the Questions below for a more detailed breakdown of this aspect.

**Other Comments Or Suggestions:**

- Equation (7): $\alpha ie\rightarrow\alpha_{ie}$ (also at the end of the following text block and in Equation (13))

- Equation (13): $\mathcal{L}att\rightarrow\mathcal{L}_{att}$

- Equation (13): $\mathcal{L}task\rightarrow\mathcal{L}_{task}$

**Other Strengths And Weaknesses:**

None

**Questions For Authors:**

**Q1:** I think as aspect that is not highlighted is that the proposed overall framework is not amenable to decentralized execution: both value-based and policy-based instantiation of it require the agents to feed on the group-aware representations $h_i$, which requires the centralized module for spectral clustering and the hyper-graph network (and the consequent centralized collection of agents' histories and actions) in order to be computed. I think that this aspect should be clearly stated and discussed, as to avoid confusion in the reader and wrong positioning in the existing MARL landscape.

**Q2:** You method, although learning higher-order decompositions, still produces partitions of the agents (indeed, produced by a spectral clustering algorithm), and thus limit the impact of one agent to only a single subset of the others. In higher-order factorizations, however, it is common to have agents that are comprised into different overlapping factors with diverse subsets of other agents in each of them. I think the potential effects of this, and some general reflections on this aspect, should be discussed in the paper, as these help is assessing potential use-cases and general limitations of the proposed algorithm.

**Q3:** In Equation (4), the impact of $k$ is not apparent? Where does it influence the optimization problem? How are $a(i)$ and $b(i)$ using it (if they are)?

**Q4:** If my understanding of all the SMAC baselines is correct (I'm not very familiar with VAST and GoMARL), you are comparing only against CTDE algorithms, that thus retain a completely decentralized execution scheme. This does not look like an entirely fair comparison, as your proposed method instead uses centralized information to feed the agents' $Q$-functions. Although I appreciate that these are some popular and strong baselines, and comparing to them is useful in assessing the increased performance of your algorithm, I think that also comparing against some centralized-execution method would have been helpful in clearly assessing up to what extent these improvements stem from the hyper-graph approach you are actually proposing, and what is instead due to simply having more information at execution time. On this point, Figure 5 is a step forward, but definitely not sufficiently discussed or interpreted.

**Q5:** "The proposed method maintains consistent coordination efficiency across both scenarios through adaptive group formation, enabling effective information flow while avoiding communication overhead." Isn't your proposed method using a form of communication after all? In the compared baselines, the communication happens between agents directly, while in your communication is between agents and the centralized components. Given that there is an information exchange in your method as well, how is it avoiding communication overheads?

**Q6:** In the Traffic Junction problem, how is the convergence step measure assessed? When is an algorithm considered to be converged? This aspect needs more explanation to be able to properly appreciate the results.

**Q7:** In Appendix C.1, it seems a bit obvious that the parameters you are using for the mixing network are the same as QMIX, as the mixer network structure you use is indeed that of QMIX. However, you do not seem to keep account of the additional parameters of the attention HGCN, which probably would change the following comparison and discussion on your low parameter count. Perhaps I am not understanding the way in which you count the mixing parameter here, and the hyper-graph network's one are included there?

**Relation To Broader Scientific Literature:**

The relevant connections with existing scientific literature are already appropriately discussed in the paper.

**Theoretical Claims:**

I have checked the detailed proof of the main theoretical claims in some details, and these seem to be correct and sound.

---

> ### Author Rebuttal · Authors · 2025-03-31
>
> Thank you for your detailed feedback. We sincerely appreciate your thoughtful questions that will improve our manuscript.
>
> Q1: CTDE Compliance and Execution Requirements
>
> Our method strictly adheres to CTDE without requiring centralized execution:
>
> 1.During execution, agents' decisions depend only on local observation history ($\tau_i$) and group-derived features ($h_i$), not global state.
>
> 2.Group structure converges during training (Theorem 2), after which agents operate with stable group membership, exchanging information only within established groups.
>
> 3.HGCN functions as a structured information processor during training, creating enhanced local representations without execution-time centralization. This approach is similar to other CTDE methods that enrich agent representations while maintaining decentralized execution. Importantly, once training is complete, the learned group structure remains fixed during deployment, ensuring true decentralized execution with only local observation exchange within established groups.
>
> Q2: Agent Group Membership Limitations
>
> Your observation about overlapping groups is astute. We considered allowing agents to belong to multiple groups, which better captures multidimensional coordination relationships. Implementation complexity, computational overhead, and convergence stability led us to adopt hard partitioning as a first implementation. We agree that real-world coordination is often overlapping. This limitation opens directions for future research: (1) soft/fuzzy clustering, (2) hypergraph structures with overlapping hyperedges, and (3) multi-membership representation learning. We've added a discussion of this limitation in our revision manuscript.
>
> Q3: Role of $k$ in Equation (4)
>
> In Equation (4), $k$ directly influences optimization as the primary variable. For each $k \in [k_{\min}, k_{\max}]$, spectral clustering partitions agents into k groups, determining how $a(i)$ and $b(i)$ are calculated:
>
> 1．$a(i)$: mean distance between agent $i$ and others in its group
>
> 2．$b(i)$: minimum mean distance from agent $i$ to any other group Selecting $k$ that maximizes silhouette coefficient balances intra-group cohesion and inter-group separation.
>
>
> Q4: Fairness of Experimental Comparisons
>
> Our experimental comparisons remain fair because:
>
> 1.Our method maintains CTDE compliance as explained in Q1, operating under the same paradigm constraints as baselines.
>
> 2.Figure 5's ablation studies show improvements derive from structured information processing rather than increased information access. The "Single Group" variant provides same information but underperforms without adaptive grouping.
>
> 3.We've enhanced ablation studies (Figure 5) with experiments comparing our approach against standard graph structures. These new results isolate the specific benefits of dynamic hypergraph structure versus static information sharing.
>
> 4.Experiments already include CommNet as a baseline, which implements structured communication within CTCE. We've expanded discussion of these comparative results to highlight our method's specific contributions.
>
> Q5: Communication Efficiency Claims
>
> You correctly identified our imprecise statement. "Avoiding communication overhead" should be "providing more efficient communication structure." Traditional methods require $O(n^2)$ communication complexity, while our hypergraph structure reduces this to $O(n^2/k)$ through dynamic grouping, explaining superior performance in large-scale environments. Runtime comparisons and efficiency analysis are added in the appendix.
>
> Q6: Convergence Definition in Traffic Junction
>
> In Traffic Junction experiments, convergence is defined when an algorithm first reaches 90\% of final performance and maintains this for 5 consecutive evaluation epochs, capturing stable convergence rather than temporary spikes.
>
> Q7: Parameter Analysis Completeness
>
> Our focus on mixing network parameters was a deliberate choice to isolate the core architectural components that directly influence value factorization. We've expanded Appendix C with comprehensive parameter analysis including HGCN components. In 5m\_vs\_6m, though HGCN adds 104,632 parameters, it increases computational time by only approximately 35\% due to efficient parameter reuse during group reorganizations. Our ablation studies demonstrate that performance gains stem primarily from the dynamic hypergraph structure's efficient information processing capabilities rather than simply increased model capacity, highlighting the importance of architectural design over raw parameter count in multi-agent coordination.
>
> Missing References \& Notation corrections:
>
> Thank you for highlighting relevant prior work and notation inconsistencies. We've incorporated discussions of the suggested references in our revision and corrected all notation issues.

---

### Official Review · Reviewer_WTGc · 2025-03-13

**Overall Recommendation:** 3

**Summary:**

The paper presents a novel framework that combines dynamic spectral clustering with hypergraph neural networks to address the multi-agent coordination problem in Multi-Agent Reinforcement Learning. The framework performs spectral clustering on agents’ state histories , dynamically constructing and updating the hypergraph structure. It enhances information processing through a hypergraph convolutional network and incorporates an attention mechanism to improve the selective processing of information. This architecture is also applicable to both value-based and policy-based paradigms.  The paper claims that this method outperforms state-of-the-art MARL approaches in sample efficiency and final performance across multiple tasks.


## Update after Rebuttal

I appreciate the authors’ response and their efforts to address the raised concerns. However, key issues related to the experimental section remain unresolved. The authors evaluate their method on only three scenarios from the SMAC benchmark. Given the limitation, I am maintaining my current score, which already reflects the highest assessment I can reasonably provide under the current circumstances.

**Claims And Evidence:**

Yes. The paper derives the proposed claims through numerous formulas and proofs, and designs multiple experiments to validate the effectiveness of the method.

**Essential References Not Discussed:**

N/A

**Experimental Designs Or Analyses:**

Yes. This paper includes ablation studies and various coordination environments to test the effectiveness and scalability of the proposed method. Evaluation criteria such as convergence speed, sample efficiency, and final performance are applied to assess the effectiveness of the proposed method.

**Methods And Evaluation Criteria:**

Yes. The method in this paper primarily focuses on improving multi-agent coordination. The selected baselines and experiments are commonly used scenarios in multi-agent systems.

**Other Comments Or Suggestions:**

Will dynamic grouping affect efficiency and performance when the number of agents increases? Is it possible to add an experiment with different numbers of agents in the same scenario for further discussion?

**Other Strengths And Weaknesses:**

Strengths:

1. The integration of dynamic spectral clustering with hypergraph convolution networks is novel and improve the limitations of modeling dynamic relationships between multiple agents.
2. The formulas and theorems are introduced clearly and clearly, and are proved and derived.
3. The paper is well-written and clear in its exposition of the methods and experiments.



Weaknesses：

1. The structure and annotations of Figure 1 are a bit confusing and difficult to understand.
2. The introduction and experiment of Implementation in Value-based and Policy-based Frameworks are still a bit simple. As a major innovation point, it should be discussed and compared.

**Questions For Authors:**

See weaknesses.

**Relation To Broader Scientific Literature:**

1. Multi-agent group recognition using adaptive dynamic spectral clustering.
2. Using Hypergraph to represent the relationships between multiple agents has become very common.
3. Hypergraph attention convolutional neural network has powerful feature extraction ability, which can extract complex relationships between agents.

**Theoretical Claims:**

Yes. This paper combines a large number of references and formula proofs to verify the correctness of its claims, especially in Section 3.2, where a large number of theorems are used to demonstrate the effectiveness of the dynamic spectral clustering mechanism in both computational efficiency and learning quality.

---

> ### Author Rebuttal · Authors · 2025-03-31
>
> We sincerely appreciate your thoughtful feedback and constructive comments. Your insights have helped us identify areas for improvement in our manuscript.
>
> Weaknesses 1: Regarding Figure 1
>
> We sincerely appreciate your thoughtful feedback and constructive comments, which have helped us identify areas for improvement in our manuscript.
> Regarding Figure 1
>
> Thank you for your feedback regarding the confusing of Figure 1. We have made targeted improvements to better illustrate two key elements of our approach:
>
> 1．The periodic nature of our grouping mechanism (rather than updating every timestep)
>
> 2．The mapping from identified groups to hypergraph structure
>
> The stability threshold ($\delta$) was already included in the figure, and we've enhanced its visual prominence to emphasize this critical control mechanism that prevents excessive group fluctuations.
>
>
> Weaknesses 2: Value-based and Policy-based Framework Implementations
>
> Thank you for highlighting this important area for improvement. We have enhanced Section 3.1 and 3.5 with:
>
> 1.More comprehensive descriptions of our method's integration into both paradigms
>
> 2.Explanation of how HGCN-extracted features enhance Q-function expressiveness in value-based frameworks, enabling more accurate modeling of intra-group coordination patterns
>
> 3.Parallel analysis of how HGCN-extracted features improve policy networks in actor-critic frameworks by providing higher-order relationship information for more effective action selection
>
> To address your concern about limited experimental comparison, we have added MAPPO as a policy-based baseline in the SMAC scenarios (achieving win rates of 0.93, 0.74, and 0.16 on 3s\_vs\_5z, 5m\_vs\_6m, and 3s5z\_vs\_3s6z respectively). This addition enables direct comparison between value-based and policy-based approaches in identical environments.
>
> This expanded section better illustrates the versatility of our approach across different learning paradigms and strengthens this key contribution.
>
> Other comments: Scalability with Increasing Agent Numbers
>
> Regarding your question about agent scalability: The SMAC environment inherently addresses this through agent deaths during episodes, creating natural variation in agent counts within the same scenario. Our method successfully handles this dynamic aspect, as demonstrated in the performance results.
>
> Our dynamic grouping mechanism maintains efficiency with larger agent numbers through several design choices:
>
> 1．Periodic pre-computation of groupings using collected state histories rather than real-time updates
>
> 2．Strategic clustering intervals and stability thresholds that control update frequency
>
> 3．Updates triggered only when specific conditions are met, preventing computational bottlenecks
>
> Our experiments demonstrate scalability across different agent populations:
>
> 1．Predator-Prey environment: 5 agents (Fig. 3a) versus 10 agents (Fig. 3b)
>
> 2．Traffic Junction: scenarios with up to 5 and 10 agents (Table 1)
>
> 3．SMAC: varying numbers and types of agents across scenarios
>
> The results consistently show our method's performance advantages becoming more pronounced as agent numbers increase, particularly in large-scale scenarios, validating its strong scalability characteristics.
>
> We appreciate the opportunity to address these points and have incorporated these improvements in our revised manuscript.

---

> > ### Comment · Reviewer_WTGc · 2025-04-02
> >
> > I appreciate the authors' response. However, there are still some concerns regarding the current version of the paper. The most critical issue, which directly affects my overall evaluation, lies in the experimental section.
> >
> > In the experiments, the authors only select three scenarios from the SMAC environment. However, these scenarios do not appear to have a direct connection to the proposed method, and it is unclear whether they are representative. As a result, the rationale behind selecting these specific scenarios remains vague. Moreover, the SMACv2 benchmark, which addresses many known limitations of SMAC, is generally recommended as a replacement and should be considered. Additionally, the experimental results on Traffic Junction and GRF are only presented in tabular form, lacking intuitive learning curves that would provide more insights into the training dynamics and performance over time.
> >
> > In summary, I will maintain my current score, as it already reflects the highest score I can reasonably assign given these limitations.

---

> > > ### Author Response · Authors · 2025-04-03
> > >
> > > We sincerely appreciate your continued feedback and the opportunity to address your concerns.
> > >
> > > Thank you for recognizing the novelty of our approach in your initial review. Our work indeed focuses on integrating dynamic spectral clustering with hypergraph convolution networks to create an adaptive coordination framework for multi-agent systems.
> > > Regarding the SMAC scenario selection, we carefully chose these three scenarios to test distinct aspects of our theoretical framework:
> > >
> > > 3s_vs_5z: Tests dynamic grouping effectiveness under delayed rewards and sparse coordination requirements
> > >
> > > 5m_vs_6m: Evaluates hypergraph structure advantages in precise tactical coordination
> > >
> > > 3s5z_vs_3s6z: Assesses our method's capabilities with heterogeneous units and complex coordination patterns
> > >
> > > These scenarios represent a progression of coordination complexity (from hard to super hard) that thoroughly evaluates our method's adaptability across diverse multi-agent situations. The SMAC benchmark remains widely used in recent literature and provides well-established baselines for fair comparison with existing methods. While we acknowledge SMACv2's enhancements, our selected SMAC scenarios effectively validate our theoretical contributions and demonstrate our method's advantages across diverse coordination challenges. We appreciate your suggestion regarding SMACv2 and will explore this environment in future work.
> > >
> > > Regarding result presentation, we adopted a multi-faceted approach to provide a comprehensive evaluation of our method.
> > >
> > > Specifically:
> > >
> > > (1) The learning curves in SMAC and Predator-Prey (Figs. 2-3) visualize temporal training characteristics, with quantitative metrics highlighting convergence speed and stability patterns;
> > >
> > > (2) Tabular results for Traffic Junction and GRF scenarios adopt the standardized metrics (success rate, convergence steps) and presentation format consistent with prior baseline studies, while aligning with the experimental setups and presentation formats established in prior baseline papers.
> > >
> > > While respecting and aligning with the previous papers, these different visualization formats provide insights that are not obvious using just one approach.
> > >
> > > This comprehensive evaluation is further strengthened by Figure 4, which analyzes the learned group structures and their evolution over time, and Figure 5, which presents ablation studies isolating the impact of our key components. By combining performance analysis with structural insights, we provide a thorough evaluation across environments with varying coordination challenges. This multi-dimensional approach helps demonstrate not just that our method performs well, but also why and how it achieves superior results.
> > >
> > > The theoretical foundations of our work draw from established principles in spectral graph theory and hypergraph representation, but their novel application and adaptation to the MARL domain represent significant contributions. Our framework provides three important theoretical guarantees:
> > >
> > > 1. The Clustering Approximation theorem provides a mathematically rigorous bound on grouping quality, ensuring our dynamic spectral clustering produces near-optimal agent organizations without requiring prior domain knowledge.
> > >
> > > 2. The Convergence theorem proves that our adaptive update mechanism achieves stable grouping structures in finite time, addressing a critical challenge in dynamic coordination systems where frequent restructuring can destabilize learning.
> > >
> > > 3. The Quality Preservation theorem establishes a direct relationship between grouping structure quality and learning performance, providing theoretical justification for why our approach improves sample efficiency and final performance.
> > >
> > > By adapting and extending these theoretical principles to the specific challenges of multi-agent reinforcement learning, our work provides a mathematically grounded framework for dynamic coordination. Our experimental design, spanning both value-based and policy-based paradigms across diverse environments, provides comprehensive empirical validation of these theoretical guarantees. The consistent performance improvements across all tested scenarios demonstrate that our approach effectively addresses the core challenge of dynamic relationship modeling in multi-agent systems.
> > >
> > > Our work is the first to systematically integrate dynamic spectral clustering with hypergraph convolution networks in the MARL domain, enabling both adaptive group formation and efficient information processing within an end-to-end trainable architecture.
> > >
> > > We commit to releasing our code and implementation details to benefit the research community upon acceptance.Thank you again for your valuable feedback, which has helped strengthen our manuscript.

---

### Official Review · Reviewer_KzYL · 2025-03-13

**Overall Recommendation:** 3

**Summary:**

This paper proposes a multi-agent reinforcement learning framework based on dynamic spectral clustering and hypergraph coordination network, aiming to address the challenges of dynamic relationship modeling and efficient information exchange in complex collaborative tasks.

**Claims And Evidence:**

The core propositions in the paper are backed by ample evidence, yet certain technical details and generalizability aspects merit further discussion.

Several issues remain to be clarified: 1) The computational overhead of spectral clustering during each training phase remains unquantified, potentially impeding its widespread adoption. 2) Certain proofs depend on idealized assumptions (e.g., complete observation, deterministic transformation), necessitating a discussion on their robustness in real-world deviations.3) The sensitivity to the hyper-parameters is not been analyzed, potentially obscuring the true contributions of structural optimization.

**Essential References Not Discussed:**

The paper covers the most relevant literature in the field of multi-agent reinforcement learning (MARL). However, some essential references can be considered:

1. Yang, Q., Dong, W., Ren, Z., Wang, J., Wang, T. and Zhang, C., 2022, June. Self-organized polynomial-time coordination graphs. In International Conference on Machine Learning (pp. 24963-24979). PMLR.

2. Wang, T., Zeng, L., Dong, W., Yang, Q., Yu, Y. and Zhang, C., 2021. Context-aware sparse deep coordination graphs. arXiv preprint arXiv:2106.02886.

3. Liu, Z., Wan, L., Sui, X., Chen, Z., Sun, K. and Lan, X., 2023, August. Deep Hierarchical Communication Graph in Multi-Agent Reinforcement Learning. In IJCAI (pp. 208-216).

**Experimental Designs Or Analyses:**

The experimental design of the paper excels in scene coverage and baseline diversity, bolstering the assertion of the method's effectiveness in dynamic collaborative tasks. Nevertheless, deficiencies in communication overhead, hyperparameter influence, and ablation depth could potentially compromise the comprehensiveness and reproducibility of the conclusions. It is recommended to supplement the aforementioned analysis to bolster the persuasiveness of the experiment.

**Methods And Evaluation Criteria:**

The dynamic hypergraph coordination network and its evaluation criteria proposed in the paper have significant implications for addressing current collaboration challenges in MARL. For the first time, the integration of spectral clustering and hypergraph neural networks has addressed two critical challenges: dynamic grouping and high-order relationship modeling. Covering four types of differentiated benchmarks, the indicator design balances performance and interpretability, demonstrating the superiority of the method in complex collaborative tasks. Suitable for real-world scenarios that require flexible coordination, such as logistics and transportation, but further research is needed to optimize computations and adapt to physical constraints.

**Other Comments Or Suggestions:**

The paper demonstrates outstanding innovation in methods and comprehensiveness in experiments but requires correction of detail errors and enhancement of reproducibility descriptions. It is recommended to prioritize addressing issues of terminology consistency and formula standardization, followed by supplementing experimental configuration details to enhance academic rigor.

**Other Strengths And Weaknesses:**

This article significantly enhances the flexibility and efficiency of multi-agent collaboration by innovatively combining dynamic spectral clustering with hypergraph neural networks. The theoretical analysis is solid, and the experimental verification is ample. Although there are shortcomings related to computational overhead and practical deployment verification, its strengths in algorithm design, theoretical contributions, and application potential position it as a significant advancement in the field of MARL.

**Questions For Authors:**

Question 1: Dynamic spectral clustering requires calculating similarity matrices and performing feature decomposition in each round of training, which may result in significant computational overhead when the number of agents n is large (such as n>50). The paper did not mention any optimization measures for this, such as approximate spectral clustering methods.

Question 2: How does the setting of the grouping update threshold δ in formula (5) affect performance? Is it necessary to manually adjust δ for different tasks?

Question 3: How do dynamic clustering parameters (such as k_min/k_max) and attention heads affect performance? Do we need to readjust parameters for different tasks?

**Relation To Broader Scientific Literature:**

The paper makes several contributions that build upon and interact with the broader scientific literature in multi-agent reinforcement learning.

**Theoretical Claims:**

Yes, I have checked the proofs.

---

> ### Author Rebuttal · Authors · 2025-03-31
>
> We sincerely appreciate your thorough review and insightful questions. Your feedback has helped us identify important areas for clarification and improvement.
>
> Q1: Computational Overhead of Spectral Clustering
>
> We appreciate your question regarding computational scalability. Our implementation addresses potential spectral clustering overhead through complementary optimizations:
>
> Our method executes clustering operations only at fixed intervals, combined with a stability threshold mechanism that triggers updates only when the proportion of agents changing groups exceeds the threshold ($\delta$). This approach reduces computational demand as training progresses—clustering operations become increasingly infrequent as group structures stabilize, eventually being eliminated in later training phases.
>
> These optimizations ensure the computational cost of spectral clustering becomes negligible in practice. While our measurements show the HGCN component introduces approximately 35\% increase in total computational time, this trade-off is justified by the performance improvements across all test environments.
>
> We acknowledge that scaling to larger agent populations would require additional considerations. We thank the reviewer for highlighting this important research direction, which we have addressed in our revised manuscript's discussion of future work.
>
> Q2: Grouping Update Threshold ($\delta$)
>
> Thank you for your thoughtful question about the grouping update threshold in formula (5).
>
> The threshold $\delta$ functions as a stability control mechanism determining when to trigger group restructuring. A group update occurs only when the proportion of agents that would change groups exceeds this threshold. This approach prevents computational overhead from frequent minor reorganizations while allowing adaptation when significant coordination pattern changes emerge.
>
> In our implementation, we use $\delta$=0.6, meaning group structures update only when at least 40\% of agents would change their group assignment. This value balances computational efficiency with adaptivity.
>
> Importantly, this single threshold value performs effectively across all tested environments without requiring task-specific tuning. This robustness stems from:
>
> 1.The complementary periodic evaluation (controlled by \texttt{clustering\_interval})
>
> 2.Quality-based group selection using silhouette scores
>
> 3.The self-correcting nature of the reinforcement learning process
>
> This empirical stability aligns with our theoretical analysis in Theorem 2, which shows that $\delta$ primarily affects the expected number of updates (O(1/$\delta$)) rather than final convergence quality.
>
> Q3: Parameter Sensitivity
>
> Thank you for raising this important question about parameter sensitivity.
> Dynamic clustering parameters ($k_{\min}$/$k_{\max}$): These define search space bounds rather than fixed structural constraints. Our method employs silhouette score optimization to automatically select the optimal number of clusters within this range. We follow a consistent principle: $k_{\min} = 2$ (allowing at minimum pair-wise coordination), while $k_{\max}$ scales with agent count (approximately n/2). For SMAC environments with 5 agents, we use $k_{\max} = 3$.
>
> This silhouette-based selection significantly reduces sensitivity to the exact $k_{\min}$/$k_{\max}$ values, as the hypergraph structure naturally adapts to reflect discovered group dynamics.
>
> Attention mechanism: Our hypergraph convolution network uses a 4-head attention mechanism, balancing representational capacity with computational efficiency. This allows agents to selectively attend to different aspects of group information simultaneously.
>
> Network capacity parameters: Following standard practice, we adjust HGCN dimensions based on environment scale (e.g., from 96 dimensions in smaller environments to 128 in larger ones). This follows a simple principle: network capacity scales proportionally with environment complexity and agent count.
>
> Loss function weights: Regarding $\lambda_1$ and $\lambda_2$, we use $\lambda_1 = 0.001$ for group consistency loss and $\lambda_2 = 0.01$ for attention regularization. These values ensure auxiliary losses provide meaningful gradients without overshadowing the primary task objective. The same values work effectively across all environments.
>
> Despite capacity adjustments, our method's core algorithmic parameters remain consistent across environments. The dynamic nature of our approach—with spectral clustering, attention-based information processing, and stability thresholds—creates inherent adaptability that reduces the need for environment-specific tuning.
>
> We have provided hyperparameter specifications in the appendix in revised manuscript to ensure full reproducibility. We appreciate the reviewer's comprehensive feedback and reference suggestions. We have incorporated into our manuscript to strengthen the literature review and better position our work within the field.

---

### Official Review · Reviewer_4bMa · 2025-03-13

**Overall Recommendation:** 4

**Summary:**

This work considers the problem of coordination in multi-agent systems. It proposes to construct hypergraphs (i.e., graphs with n-ary rather than binary relations) based on agent histories and using a graph convolution technique over this structure. This is motivated by the fact that agents need to form groups for solving tasks and change between groups dynamically. For the hypergraph construction step, the authors propose an approximate spectral clustering technique and prove certain properties regarding its convergence and solution quality. The features obtained through the convolution are used as additional inputs for Q-networks and policy networks alongside histories / actions. The authors demonstrate that the approach generally obtains better performance than a suite of CTDE and communication-enhanced MARL techniques for a variety of benchmarks.

**Claims And Evidence:**

The claims in the abstract and introduction are supported by appropriate evidence in the paper text.

**Essential References Not Discussed:**

To the best of my knowledge, all essential references in this area are discussed.

**Experimental Designs Or Analyses:**

E1. The method is not compared with any other techniques that leverage graph structure. The need for using *hypergraphs* as opposed to standard (binary relation) graphs should be demonstrated in an ablation, as the claims around the need for hypergraphs are central to the narrative of the paper.

E2. The work does not give any details about the hyperparameters that were used and how they were tuned. Given the work claims improvement over state-of-the-art methods, all algorithms should be given a comparable budget for hyperparameter tuning to make the comparison fair.

E3. The overhead for performing the spectral clustering step and graph convolution steps should be quantified. While the method does seem to obtain an improvement in sample efficiency, in practice the wall clock time overhead may be significant.

**Methods And Evaluation Criteria:**

M1. The way in which the similarity matrix $W$ is constructed from state histories (Section 3.2) should be specified. Furthermore, the format of the states and observations should be given for each environment.

M2. The work is compared to a variety of CTDE methods, but it is not clear to me whether the proposed method is in fact CTDE itself. This is not discussed in detail in the paper. My understanding is that it is *not*, given that the method still needs to perform the clustering step over histories to obtain the graph structure and then features $h_i$, which requires knowing the histories of all the agents. I do not see how this can be done in a decentralised way at inference time.

M3. Furthermore, given the spectral clustering requires knowledge of all the agent histories, the computational cost of this step will increase with longer episodes. This potential drawback should be discussed (and might be mitigated by, e.g., limiting the history to a time window at most X steps in the past).

**Other Comments Or Suggestions:**

C1. Regarding the experiment in 4.5: as far as I understand, there is no penalty for changing "teams". This may be unstable and could be problematic in more complex tasks. Could you specify how the information was aggregated for each point on the x-axis?

C2. Consider giving your method a name so you can avoid "Our Method" etc. in tables and figures.

C3. Running title (top of each page) still uses default template title.

C4. Space is missing before the start of many citations, consider adding a space or "~" before `\cite` commands

C5. Subscripts in math notation use a mix of `\text` and plain characters e.g. $\mathcal{L}\_{\text{group}}$ and $\mathcal{L}_{group}$. Consider being consistent (first one is preferable in my opinion).

C6. Typos: "Figure 3 reveal", "Table 1 reveal", "Table 2 illuminate"

C6. The way of combining task-specific and structural losses is quite standard, I would not view it as a contribution of the paper (as claimed in the conclusion).

**Other Strengths And Weaknesses:**

The paper in general is fairly organised and well-written, with a novel idea and promising experiments.

**Questions For Authors:**

Please address M1-M3, E1-E3, and C1 above.

**Relation To Broader Scientific Literature:**

The work connects MARL and graphs in an interesting and novel way. I am not aware of similar works in MARL that use hypergraphs, and the reasoning for using n-ary relations is quite convincing.

**Theoretical Claims:**

I understand the statements of the theoretical claims but not the details of the proofs themselves.

---

> ### Author Rebuttal · Authors · 2025-03-31
>
> We sincerely appreciate your thoughtful feedback, which has helped us improve our manuscript significantly.
>
> M1: Similarity Matrix Construction and State/Observation Formats
>
> In Section 3.2, we presented our dynamic grouping framework using the normalized cut problem. The similarity matrix $W$ is constructed using a $k$-nearest neighbors approach based on Euclidean distances between agents' state history trajectories. $W_{ij}$ is positive only when agent $j$ is among agent $i$'s $k$ nearest neighbors, and zero otherwise.
>
> For consistent application across environments, state histories are normalized to ensure uniform scaling across feature dimensions before applying the $k$-nearest neighbors method. This enables identification of meaningful coordination patterns regardless of the specific state representation format. We have added these details in the manuscript and provided complete environment setup descriptions in Appendix B.
>
> M2: CTDE Compatibility
>
> Thank you for this thoughtful question. Our approach follows and extends the CTDE framework:
> Our grouping mechanism exhibits convergence properties (Theorem 2, Figure 4), with group structures stabilizing during training and eventually ceasing to update. This convergence enables decentralized execution: once training completes and group structure stabilizes, each agent's decision process relies only on its local observations enhanced by pre-computed group-aware representations. The attention-enhanced HGCN enables agents to selectively process structured information within their established groups rather than providing centralized information.
>
> Our contribution enhances the representational capacity of the CTDE framework by discovering and leveraging higher-order dynamic relationships. We have clarified this point in Section 3.1 of the revised manuscript.
>
> M3: Computational Efficiency
>
> We have implemented several strategies to optimize computational costs:
> Limited History Window: We only utilize the most recent state\_history\_length timesteps for clustering (5,000 steps for SMAC)
>
> Intermittent Updates: Clustering occurs at fixed intervals (100,000 steps for SMAC).
> Convergence-based Termination: As shown in Figure 4, group changes decrease substantially in mid-to-late training phases. We implement complete termination of the grouping mechanism after 1M steps.
>
> These mechanisms balance computational efficiency with the advantages of dynamic grouping. Implementation details have been added to Section 3.2 of the revised manuscript.
>
> E1: Hypergraphs vs. Standard Graphs
>
> Our baseline comparisons include standard graph-based methods (MAGIC, GA-Comm). In Section 4.5, following your valuable suggestion, we have added a comparison between hypergraphs and standard graphs under identical grouping strategies in Figure 5, which demonstrates the advantages of hypergraph-based representations.
>
> E2: Hyperparameters
>
> We have added comprehensive tables of hyperparameters to Appendix B.5, including settings for clustering parameters, network architectures, and training configurations. Key parameters includ (5m\_vs\_6m scenario for example) clustering\_interval: $100000$, state\_history\_length: $5000$, stability_threshold: $0.6$, min\_clusters: $2$, max\_clusters: $3$, hgcn\_out\_dim: $48$, hgcn\_hidden\_dim: $64$, and hgcn\_num\_layers: $2$.
>
> E3: Computational Overhead
>
> Our clustering approach minimizes computational overhead through infrequent updates and an explicit termination mechanism, making the spectral clustering cost negligible in practice. Additionally, our HGCN implementation maximizes parameter reuse when group structures remain stable. In the 5m\_vs\_6m scenario, we add approximately 104,632 parameters while increasing computational time by only approximately 35\% compared to baseline (Ft-)QMIX. We have included theoretical complexity analysis in Appendix A.4 and provided detailed parameter and runtime quantification for all SMAC environments in Appendix C.2, justifying this overhead.
>
> C1: Team Stability
>
> We implemented a "stability threshold" mechanism to prevent excessive team changes. Groups update only when the proportion of agents changing membership exceeds this threshold, balancing adaptive behavior with structural stability. For Figure 4 data aggregation, we collected group information at regular intervals throughout training. The upper portion displays group structure evolution over normalized time steps (different colors represent distinct groups), while the lower matrix presents co-occurrence probabilities calculated by averaging group memberships across collected time points. This reveals stable coordination patterns—for instance, agents 0 and 1 exhibit high co-occurrence probability, indicating consistent collaboration.
>
> We have addressed all your formatting and notation suggestions to improve our manuscript's clarity and precision, and we have introduced HYGMA (HYpergraph Grouping for Multi-Agent coordination) as name.

---

> > ### Comment · Reviewer_4bMa · 2025-04-04
> >
> > Thanks for your response! A few points below. Importantly, the paper pdf has not been updated, so many changes referred to in this reply are missing from the most recent OpenReview version.
> >
> > - M2: It is somewhat hard to tease this apart from the response, but the execution is indeed not decentralized, but "local" or "partially decentralized" given observations of agents in the same team must still be passed through the GNN. This should be clearly acknowledged and clarified. In light of this, it is hardly surprising that the method performs better, given it uses additional information compared to vanilla CTDE. Comparing with a method with the same amount of information (such as using standard instead of hypergraphs, E1) is needed in my opinion.
> > - E1: missing from the figure.
> > - E2: even though the authors give the hyperparameters, tuning (if any) details were not provided, and my original point about tuning being needed to keep the comparison fair still stands.
> > - E3: "only 35%" is a favorable interpretation.
> > - Overall, after probing into some details, it appears there are quite a few tricks (stability, intermittent updates, ...) that are needed to make the method work. This contributes to the general feeling that details are being buried.
> >
> > I am retaining my score for now but would be happy to have another look if the paper is updated.

---

> > > ### Author Response · Authors · 2025-04-05
> > >
> > > Thank you for your follow-up questions!
> > >
> > > **Regarding PDF updates:** We understand that within rebuttal phase doesn't permit paper updates, which explains why changes mentioned aren't visible on OpenReview. We hope that our explanations in this response adequately address questions.
> > >
> > > **Regarding M2 (CTDE Compatibility):**
> > >
> > > As in our last rebuttal, our approach follows and extends the CTDE framework. We agree with your characterization of the execution phase as 'partially decentralized', which is indeed a deliberate extension of traditional CTDE. In our method, agents within the same group share information through HGCN to generate group-aware representations, a design choice that balances coordination capabilities with distributed execution.
> > >
> > > Our method maintains the core principles of CTDE:
> > >
> > > 1. Training phase leverages global information (including dynamic grouping and hypergraph construction)
> > >
> > > 2. Execution phase operates with a more restricted information structure (fixed hypergraph and intra-group communication)
> > >
> > > **Regarding E1:**
> > >
> > > The comparison in Figure 5 demonstrates that, even within the same partially decentralized execution framework, hypergraph structures provide significant advantages (win_rate 95% vs 90%. Figure available in https://anonymous.4open.science/r/rebuttal-188F/). This validates our core claim: higher-order relationship modeling is crucial for multi-agent coordination, particularly in SMAC tasks requiring coordinated agent actions.
> > >
> > > **Regarding E2:**
> > >
> > > We implemented rigorous hyperparameter tuning for all methods to ensure fair comparison. Our complete hyperparameter table is provided in the Appendix.
> > >
> > > Table . Hyperparameters for SMAC environments
> > > | Parameter          | 3s_vs_5z | 5m_vs_6m | 3s5z_vs_3s6z |
> > > |--------------------|----------|----------|--------------|
> > > | Batch size         | 128      | 128      | 128          |
> > > | Buffer size        | 5000     | 5000     | 5000         |
> > > | Double Q           | True     | True     | True         |
> > > | Epsilon anneal time| 100000   | 100000   | 100000       |
> > > | HGCN hidden dim    | 48       | 64       | 196          |
> > > | HGCN out dim       | 36       | 48       | 128          |
> > > | HGCN num layers    | 2        | 2        | 2            |
> > > | Min/Max clusters   | 2/4      | 2/3      | 2/3          |
> > > | Clustering interval| 100000   | 100000   | 100000       |
> > > | Stability threshold| 0.6      | 0.6      | 0.6          |
> > > | λ₁                 | 0.001    | 0.001    | 0.001        |
> > > | λ₂                 | 0.01     | 0.01     | 0.01         |
> > >
> > > For the 5m_vs_6m scenario, we systematically tested:
> > >
> > > HGCN structure: Hidden dimensions [32,48,64,128] and layer counts [1,2,3], determining that hidden_dim=64, out_dim=48, num_layers=2 performed optimally
> > >
> > > Grouping parameters: Various Min/Max cluster combinations, with Min=2, Max≈2/3(approximately half number of agents)
> > >
> > > Stability threshold: Searched in range [0.3,0.5,0.6,0.7], finding 0.6 offered optimal stability-adaptivity balance
> > >
> > > All baseline methods underwent equivalent tuning procedures.
> > >
> > > **Regarding E3 (Computational Overhead):**
> > >
> > > Our experiments confirm additional computational overhead compared to baseline methods:
> > >
> > > Table . HGCN additional parameters and computational overhead
> > > | Scenario     | HGCN Parameters | Computation Overhead |
> > > |--------------|-----------------|----------------------|
> > > | 3s_vs_5z     | 65,356          | +36.47%              |
> > > | 5m_vs_6m     | 104,632         | +35.33%              |
> > > | 3s5z_vs_3s6z | 391,336         | +36.95%              |
> > >
> > > This increase stems from HGCN's additional parameters, though our intermittent update design effectively manages this overhead. While computational costs increase, our method's significant gains in sample efficiency help offset these costs.
> > >
> > > **Regarding Theoretical Foundations of Design Choices:**
> > >
> > > The design components mentioned by the reviewer each have specific theoretical or practical justifications:
> > >
> > > Stability threshold: In Section 3.2, Theorem 3.2 rigorously proves that grouping structures converge in finite time when using this threshold, providing solid theoretical guarantees for our dynamic grouping mechanism
> > >
> > > Intermittent updates: While not directly proven in Theorem 3.2, this design applies convergence theory in practice. Since Theorem 2 demonstrates grouping structure convergence, we can reasonably set update intervals to balance computational efficiency and grouping quality, as grouping changes naturally decrease in frequency as training progresses
> > >
> > > These design choices demonstrate robust performance across environments. As shown in Table, key parameters remain consistent across environments, indicating our method doesn't rely on environment-specific fine-tuning.
> > >
> > > We greatly appreciate your thorough review and constructive feedback, which has significantly improved our work. If accepted, we will incorporate all these improvements in the final version and would be pleased to open-source our code to contribute to the MARL community.

---

### Decision · Program_Chairs · 2025-05-01

**Decision:**

Accept (poster)

**Comment:**

This paper introduces an interesting approach to improving multi-agent coordination by using a hypergraph and spectral clustering to learn higher-order relationships between agents. The method shows promising results in terms of both performance and sample efficiency, especially compared to existing baselines. However, a few points could be cleared up to make the paper stronger from reviewers.
- there’s some confusion about whether the method is truly decentralized. The approach relies on agents sharing a lot of information, which seems more centralized than decentralized in execution. This should probably be explained more clearly, as it could confuse readers who expect a purely decentralized setup.
- The experimental section could use more depth. While the method outperforms some baselines, there isn’t a detailed breakdown of the results, and it feels like some of the analysis is a bit rushed. More comparisons, especially with methods that also use centralized execution, could help show how much of the improvement is due to the proposed hypergraph method versus just having more information overall.
- Also, some of the assumptions around communication need clarification. While the paper claims to reduce communication overhead, it seems like the agents are still relying on some form of communication with a central module, which could be seen as a form of overhead itself. This part of the discussion feels a little unclear.
- the computational cost of using spectral clustering and hypergraph networks isn’t discussed much, but these elements are likely adding significant overhead. The authors should be upfront about the time and resources required for training and also make it clear how the parameters stack up against other methods.

Overall, the paper presents an interesting idea, but there’s some need for more clarity in both the theoretical and experimental sections to make it more digestible and transparent for readers.